# A comprehensive scan of psychological disciplines through self-identification on Google Scholar: Relative endorsement, topical coverage, and publication patterns

Jaap J. A. Denissen[1]*, John F. Rauthmann[2]

1 Department of Psychology, Utrecht University, Utrecht, Netherlands, 2 Department of Psychology, LMU München, Munich, Germany

* jjadenissen@gmail.com

**Data Availability Statement:** We share all data and code of this project on our Open Science Framework page https://osf.io/rj9ae/?view_only=022e120070514a748f7a3dab07dfefb8.

## Abstract

Psychological researchers often identify with psychological disciplines, such as social or clinical psychology. The current study analyzed Google Scholar profiles from 6,532 international scientists who attracted more than 100 citations in 2019 and self-identified with at least one of 10 common psychological disciplines (psychoanalysis; clinical psychology; (cognitive) neuroscience; developmental psychology; educational psychology; experimental psychology; biological psychology/psychophysiology; mathematical psychology/psychometrics; social psychology; personality psychology). Results indicated that almost half of all psychologists self-identified with either social psychology or cognitive neuroscience. There were 487 topics that were endorsed at least five times, ranging from highly discipline-specific topics to more integrative ones, such as *emotion* and *personality*. We also factor-analyzed frequencies of topical endorsement across disciplines and found two factors, which we interpreted as reflecting correlational and experimental research traditions (with social psychology being the largest discipline within the former tradition and cognitive neuroscience being the largest discipline within the latter tradition). Differences in productivity and impact were also found, with researchers identifying with psychometrics being the most productive and researchers identifying with personality psychology, cognitive neuroscience, and multidisciplinary psychology as the most impactful in terms of citation increases per additional output. Recommendations for promoting cross-fertilization across psychological disciplines are formulated.

## Introduction

Soon after the emergence of psychology as an independent field of research in the second half of the 19th century, different disciplines have emerged within it [1, 2]. Scientists have divided themselves further into more circumscribed disciplines, such as developmental and social psychology. These divisions have often been criticized because they reduce cross-talk between researchers and lead to isolated research efforts that would benefit from a multidisciplinary

**Funding:** The authors received no specific funding for this work.

**Competing interests:** The authors have declared that no competing interests exist.

perspective [3]. For example, one risk is that members of segregated disciplines are no longer informed about the topics of other disciplines [4]. Also, without much direct contact, it becomes difficult to compare disciplines, let alone integrate their respective insights. The current research set out to conduct a systematic scan of Google Scholar (GS) profiles. Although this methodology is less suited to inform conclusions about disciplinary formation or the social organization of scientists, GS profiles do offer unique insights into the interface between individual researchers and their disciplinary identification. By investigating correlates of these disciplinary identifications, tentative conclusions might be drawn about the commonalities and uniquenesses of major psychological disciplines. We did so by investigating three basic research questions, comparing psychological disciplines in 1) their relative endorsement across time and world regions, 2) their topical coverage, and 3) publication patterns.

## Classification of psychological disciplines

Various systems exist to classify psychological disciplines, using, for example, empirical or rational arguments [1]. Some approaches have focused on fundamental theoretical paradigms and identified "schools" within psychology. For example, Robins et al. [5] identified four schools: Psychoanalysis, behavioral psychology, cognitive psychology, and neuroscience. In the Netherlands, Duijker [6] introduced five basic disciplines: experimental psychology, methods and statistics, developmental psychology, personality psychology, and social psychology. In Germany, the German Psychological Society has declared several fundamental disciplines as part of its curriculum framework, overlapping with Duijker's but also including biological psychology and more applied disciplines such as psychological assessment, clinical psychology, educational psychology, and industrial/organizational psychology [7]. Finally, the widely used Web of Science database currently distinguishes 10 psychological disciplines, which are comparable with the German classification, with the exception that psychoanalysis is included in Web of Science, whereas personality psychology is not.

For the present study, we adopted the Web of Science classification because it is often used in bibliometric research (e.g., [8]). However, we decided to modify it in a number of ways. First, we added personality psychology because it is considered a core psychological domain in many countries (e.g., in the Netherlands and Germany). Second, we did not adopt the Web of Science domain "applied psychology" as a discipline because a) we deemed it too heterogeneous to be useful (i.e., there are many ways to "apply" psychology, for example, in forensic settings, work and organizations settings, etc.) and b) there were too few psychologists ($< 100$) who endorsed this label in GS. Third, we added cognitive neuroscience as a separate field because of a) its increasing prominence [5] and b) its inclusion in Web of Science as a separate interdisciplinary category of "neurosciences". Fourth, we did not include the category "multi-disciplinary psychology" because a) we intended to create this category empirically (see below) and b) this is not a common term psychologists self-identify with. In total, we ended up with 10 disciplines: psychoanalysis, clinical psychology, cognitive neuroscience, developmental psychology, educational psychology, experimental psychology, biological psychology, mathematical psychology, social psychology, and personality psychology. We used this categorization of 10 psychological disciplines to address three research questions, which we outline in the following sections.

## Research questions

**Relative endorsement and international representation of disciplines.** Our first research question pertains to the relative endorsement of psychological disciplines across time and countries. The relative endorsement of psychological disciplines is defined as the relative

percentage of psychological scientists that identify with a particular psychological discipline (as expressed in GS profiles). Robins, Gosling, and Craik [5] studied changes in four schools in psychology: psychoanalytic, behavioral, cognitive, and neuroscience. They observed that the predominance of cognitive perspectives increased sharply during the 1970s, as measured by increasing use of corresponding keywords in articles and dissertations as well as a relative increase in the number of citations. Their analysis also showed a decline in prominence of psychoanalysis, and an unexpected lack of increase in prominence of the (then still rather nascent) neuroscience school. Note that the "schools" as studied by Robins et al. [5] represent paradigms that can be theoretically applied to many (if not all) disciplines. For example, the cognitive paradigm can be applied to developmental psychology, social psychology, education psychology, and so forth. In contrast, disciplines additionally have organizational features, such as its own journals, conferences, and scientific associations, and oftentimes they are also reflected in institutional structures, such as the formation of separate units (e.g., departments). The current study investigates disciplines in this latter regard, although we acknowledge that it can sometimes be difficult to separate disciplines from paradigms (e.g., in the case of "experimental psychology").

Another focus of our analysis is on the relative prominence of Anglo-Saxon countries like the US in psychological research. There have been frequent criticisms of the overreliance of Western samples in psychological research [9, 10], but it is equally important that there is diversity in terms of authors' cultural and ethnic background–not just within any diverse country (e.g., the US, Brazil, South Africa, etc.), but also between countries. Comparing the share of US contributions over time, a relative decrease over time has been reported both for the period between 1975 to 1994 [11] and between 1996 to 2010 [12]. To the best of our knowledge, however, no systematic analysis has focused on relative differences between world regions in the relative identification with psychological disciplines. It has been argued that cultural and economic background is relevant for the relative endorsement of certain scientific paradigms [13]. For example, it has been argued that the drive model underlying psychoanalysis is a typically individualistic model that does not sufficiently take relational considerations into account [14]. Whether there are indeed systematic differences between countries in their researchers' self-identification with psychological disciplines is still an open question, however.

**Topical coverage of psychological disciplines.** Our second research question pertains to differences between psychological disciplines in the kinds of topics that are specific to each discipline. We did not have firm predictions. Some domain-specific topics seemed obvious, for example, that mathematical psychology would be focused most on statistical techniques (e.g., multilevel modeling, structural equation modeling, etc.). Other than that, linkages of topics to disciplines is a complex endeavor and often depends more on established traditions than on logical classifications. For example, "attention" or "perception" are common topics for experimental disciplines, such as experimental psychology and neuroscience, but they could theoretically be also investigated in other disciplines (e.g., development of perception in children; attention bias in clinical psychology). Because of this, we did not derive firm predictions regarding the topics that would emerge as discipline-specific. For the same reason, we did not have good reasons to expect certain topics to be more multidisciplinary than others. The only exception was the discipline of personality psychology, which has been identified (Yang and Chiu actually based their conclusion on the hub position of the *Journal of Personality and Social Psychology*, which is (by its name) a mixed journal. However, their subsequent interpretation relied most directly on personality psychology as a unifying discipline that studies the "whole person".) by Yang and Chiu [15] as a so-called "hub science"–a discipline that produces knowledge that is widely used by other disciplines. It might thus be expected that topics that

are often studied in personality psychology might have a greater multidisciplinary appeal. Of note, however, Yang and Chiu used citations patterns related to APA flagship journals of different psychological disciplines, whereas we used citation patterns linked to individual researchers self-identifying with such disciplines, so it is unclear whether their results would generalize to the current study.

Finally, we investigated whether the relative emphasis by researchers on certain research topics would resemble established meta-distinctions between psychological traditions. Cronbach [16] has already remarked that psychology can be divided by approaches that use a correlational methodology versus approaches that use an experimental methodology. In correlational psychology, differences in people's everyday behavior are investigated, oftentimes using survey methodology or observations. In experimental psychology, general processes are studied in controlled laboratory settings, oftentimes using reaction times and physiological signals as indicators (for an overview of distinctive features, see [17]). This broader distinction was recently validated in an empirical analysis of words appearing in abstracts of Dutch psychological articles: Clear "continents" (i.e., spatially clustered groups) of correlational versus experimental terms emerged [18]. We thus expected that we would also find evidence for such distinctions in an analysis of research topics as endorsed in international GS profiles.

**Publication patterns of psychological disciplines.**    Our third research question pertains to differences between disciplines in productivity and citation impact. Comparing the impact of different psychological disciplines can be useful if institutions must decide which psychological discipline to invest in or to devise strategies to achieve the most impact. Also, when comparing researchers from different psychological disciplines, it is important to know the average benchmark of these researchers' disciplines to compare their *relative* performances. There are various sources of impact differences between disciplines. In the following, we discuss two of them: the centrality of the discipline and the robustness of its findings, though others might also apply.

Regarding the former, when a discipline is a hub science, it receives citations from many other disciplines and accumulates more impact than disciplines that are more at the periphery of the discipline citation network. As stated above, Yang and Chiu [15] identified personality psychology as a hub science, which might translate to more impact for scientists who identify with that discipline (see also [19]). Regarding the latter, disciplines might differ in the replicability of their findings. In a widely cited analysis [20], for example, cognitive psychology studies were found to be on average across several effects more frequently replicable than social psychology studies [21]. In addition, findings from personality psychology have recently been identified as especially likely to replicate [22].

## Methodological issues in comparing disciplines

There are multiple ways to compare the impact of disciplines, each with their advantages and disadvantages: Using journal impact metrics, attending to institutional or organizational structures, or crawling publicly available author profiles. In the following, we will compare these different approaches.

One way is to look at the average (or median) impact factors of journals that are associated with a discipline. For example, the Journal Citation Report [JCR; 23] identifies 9 separate psychological categories and provides information about the average impact factor within these categories. For psychology, journal impact in the neurosciences is thus determined to be highest, and impact in psychoanalysis the lowest. Besides being frequently criticized as a problematic indicator of scholarly quality (e.g., [24]), a disadvantage of relying on the impact factor is that journal classification systems are typically domain-general, so their disciplinary divisions

can appear somewhat haphazard. For example, JCR derives most categories from subject matter (e.g., social psychology, developmental psychology) but also includes some categories based on methodology (experimental psychology) or theoretical approach (psychoanalysis). Furthermore, a representative scholar of certain disciplines might publish in cross-domain outlets (e.g., *Psychological Review* or *PNAS*) or in outlets of other disciplines (e.g., a personality psychologist publishing in *Educational Psychology*). This hampers a comparison between disciplines if journal classifications are used.

Impact across disciplines can also be compared by establishing individual indices from representative groups of researchers, such as within a certain university or scientific associations. For example, representatives of disciplines might be identified by means of their faculty affiliation (e.g., Department of Clinical Psychology). A potential problem, however, is that many universities do not have organizational structures that mirror the disciplinary organization of psychology. For example, many universities do not have a department of educational or personality psychology, although these are clearly recognizable sub-disciplines. Representatives of certain disciplines might also be drawn from lists of editorial boards of prominent disciplinary journals, or from the boards of learned associations. This has the disadvantage that only relatively prominent researchers are sampled, which would not allow for a fair comparison across different career stages.

A third approach, which was adopted in the current paper, is to use researchers' (self-)identified disciplines on a publicly available bibliometric search engine, such as Google Scholar (GS). This bibliometric resource was launched in 2004 and is free, popular, and widely used by psychological scientists today–and is therefore often preferred because it also captures non-journal publication outlets that are relevant for some disciplines but not others, such as conference proceedings [25]. Since 2011, it is possible for researchers to create a profile that lists their contribution, and also list "areas of interest", which are typically used to specify the researcher's sub-discipline and/or topics of interest. Using self-identified sub-disciplines in GS has a number of key advantages. For example, it allows for researchers to describe up to five research topics in their own words, thus minimizing artificial or otherwise biased categorizations. The fact that multiple topics are possible also allows researchers to identify with more than one discipline. The GS scholar database also assigns a unique ID to each researcher, thus allowing longitudinal analysis of productivity and citation patterns. Finally, scholars of all career stages and backgrounds can create profiles on the platform, which increases the diversity of the overall pool of researchers that might endorse one of the targeted psychological disciplines. That said, the use of GS also has a number of important limitations, such as the fact that not all researchers have GS accounts, not all GS accounts specify one or more research topics, and not all specified topics can easily be assigned to one of the selected psychological sub-disciplines. We aimed to partially address these limitations through some exploratory analyses and will revisit them in the Discussion section.

## The current study

The current paper used researchers' self-endorsed identifications with psychological disciplines as a starting point of a comprehensive scan of public profiles of psychological scientists. We used GS to identify researchers by means of labels related to 10 major psychological disciplines (psychoanalysis; clinical psychology; (cognitive) neuroscience; developmental psychology; educational psychology; experimental psychology; biological psychology/psychophysiology; mathematical psychology/psychometrics; social psychology; personality psychology). The researchers' profile and citation data were then used to address three research questions.

For *Research Question 1*, we investigated distributions of self-endorsed disciplines so that we could empirically establish the relative frequency of researcher profiles with a multidisciplinary background as well as changes in prominence of the different disciplines over time. For *Research Question 2*, we looked at self-endorsed labels of all profiles to identify topics that are characteristic for certain (groups of) disciplines, as well as topics that are highly cross-disciplinary. For *Research Question 3*, we compared the impact of psychological disciplines, both in terms of average productivity per year as well as cumulative citation impact. To address these questions, we used the GS profiles to create average findings for the 10 psychological disciplines across hundreds of scholars each.

Our approach has several features that set it apart from other literature. First, we took a broad approach focusing on all psychological disciplines but also went into depth regarding one discipline that is often left out of analyses: personality psychology. Furthermore, we used Google Scholar to compare disciplines, which has not been done before but has several advantages. For example, it allowed us to flesh out the topics that each discipline tackles and also to identify topics that are covered by multiple disciplines. Our findings can thus give rise to more constructive suggestions for topics that have the most potential for interdisciplinary collaboration. Furthermore, our method allows for the identification of linkages with the individual scholar as unit of analysis, so novel links can emerge (e.g., if topics covary in scholarly interest profiles but are typically investigated in separate papers). Finally, our analysis covers a broader timespan that has featured many important developments, for example, the increased globalization of academic scholarship. Our method also has important limitations, however, which we cover in the Discussion section.

## Method

We share all data and code of this project on our Open Science Framework page https://osf.io/rj9ae/?view_only=022e120070514a748f7a3dab07dfefb8 so that others can reproduce our analyses. Because only public data were used, no ethical permission was deemed necessary. For privacy reasons, we refrained from sharing identifying information (such as GS identifiers, scientists' given names, or the labels they endorsed) in our uploaded materials but all findings can still be reproduced with the shared materials.

### Procedure

**Extraction and processing of profiles.**   For each of the 10 focal disciplines, GS profiles were identified by using the "label" function in searches. For example, for social psychology, "label: social_psychology" was entered in the search bar to identify profiles. Search results are displayed by GS in groups of ten, in descending order of the total number of citations for each researcher. We saved each page until the point when the number of citations per scholar on the corresponding results page became lower than 100. For "mathematical psychology" and "biological psychology", the number of result pages was lower than 10, suggesting that scientists from these disciplines might perhaps use different terminology to identify their domain. By inspecting the journal titles from these categories in the Journal Citation Reports (JCR), we identified "psychometrics" and "psychophysiology" as alternative labels, which indeed produced sufficient results in GS. For neuroscience, we added the suffix "cognitive", to ensure sampling of psychological researchers (in contrast to, for example, medical neuroscientists) and because the label "cognitive neuroscience" is frequently used in combination (e.g., as in the Cognitive Neuroscience Society).

**Crawling of citations and refinement of data.**   We used automatized scripts to extract career information of each GS profiles, using the "comparison" function of the scholar R-

package "scholar", Version 0.1.5. [26]. This package extracts the number of citations for each "career year" of different scholars. The crawling for all disciplines took place in February of 2019. Because of this, we only included citation data up until 2018. For every included GS identifier, the crawling produced one row per year in which the scholar was cited (i.e., long format data), with the number of citations and the "career year" (0 in the year of the first publication, 1 one year after the publication, etc.) of the scholar as additional columns.

Further inspection indicated that it was beneficial to compute the year of first publication by hand, in addition to relying on the scholar package. This had two reasons: a) citations are only tracked from 1980 onwards in GS and b) the "scholar" package relies on the first bar of the "citations per year" graph although this method sometimes produces incorrect data. Further inspection indicated that the latter primarily happened when profiles from scholars (especially with common names) included publications from multiple scholars. Furthermore, this happened for some highly prominent and established researchers, for whom the citation counter started much later than the year of their actual first publication. We solved this by implementing an algorithm that extracted the first publication from the total publication output list for a researcher, looking for the year of the first publication that sufficed three criteria: a) featuring the author's last name in the list of authors, b) attracting at least 10 citations, and c) being followed by another publication within 3 years. To test the performance of this algorithm, we manually inspected 106 cases in which the discrepancy between the scholar estimate and our own algorithm exceeded 15 years. This inspection confirmed the validity of our algorithm. Because our algorithm relied on the last name of the author (see Criterion a), it did not work for 709 profiles (mostly due to naming issues, for example, when researchers added a suffix such as "PhD" to their name or included foreign characters in their name). For these profiles, we used the estimate produced by the scholar package, which was still justified: When predicting the classic GS estimate with our algorithmic alternative, the association was extremely high, $\beta = .91$, $p < .001$.

Second, citation rates might depend on the quantity of papers. Accordingly, we extracted the number of rows resulting from the "get_publications" function of the "scholar" package. To transform this into a score of productivity, this number was divided by the difference between the author's year of last publication and the author's year of first publication. Because this variable had some extreme outliers, the maximum productivity score was capped at 40.

Third, inspection of first results indicated that a non-trivial number of scholars participated in the multi-author publication about the replicability of psychological findings [19]. This publication greatly inflated the citation count of 253 authors, many of them relatively junior, so the paper had the potential to strongly bias career progression estimates. Because participation in this paper was skewed across disciplines (with an over-representation from social psychology and some other disciplines), it seemed necessary to correct for this. Accordingly, we flagged scholars who were included in the author list, amounting to 46 scholars, and excluded their citations to the replication article.

## Sample

**Number of profiles.** We obtained 6,880 profiles by crawling GS using the above-described keywords. Table 1 shows the total number of profiles for each of the 10 disciplines. Strikingly, cognitive neuroscience (29%) and social psychology (23%) were by far the largest disciplines. After merging duplicate profiles, we ended up with information about 6,532 researchers.

**Geographical spread of profiles.** GS encourages users to verify their email address, of which only the domain name is displayed in the public profiles. This information was provided by all but 270 scholars. In addition, 248 profiles were linked to an ".org" or ".com" email address,

**Table 1. Disciplines and their numeric strength.**

| Discipline name | Number of profiles | Percentage multidisciplinary |
|---|---|---|
| Psychoanalysis | 190 | 9% |
| Clinical psychology | 900 | 9% |
| Cognitive neuroscience | 1,920 | 5% |
| Developmental psychology | 670 | 15% |
| Educational psychology | 420 | 14% |
| Experimental psychology | 230 | 25% |
| Personality psychology | 120 | 43% |
| Psychophysiology | 350 | 14% |
| Psychometrics | 610 | 7% |
| Social psychology | 1,470 | 8% |
| **Merged total** | **6,532** | **5%** |

*Note.* The numbers do not add up to the merged total number because profiles that endorsed multiple profiles were counted only once in the final tally. Also, the percentage of multidisciplinary profiles is lower than the average percentage across disciplines because of a) large differences in the size of disciplines and b) the fact that multidisciplinary profiles are counted in multiple disciplines but only appear once in the merged dataset (i.e., if all multidisciplinary profiles would endorse only two disciplines, then the average total percentage would be only half of the percentage in each discipline).

which could not be used as an indicator of geography. We used the remaining extensions as a proxy for the country of the authors' primary affiliations. The resulting extensions were tabulated according to the email extension, which equaled the country code (e.g., ".fr" for France) except in one case: The ".edu" extension can theoretically be used by institutions in all countries but is predominantly used by US institutions. Results indicated that the USA (as indicated by the.edu extension) was by far the country with most author profiles, accounting for more than a third of all profiles. Furthermore, more than 50% of all profiles belonged to one of four Anglo-Saxon countries: The USA, the UK, Canada, and Australia. In fact, these countries occupied four spots in the Top 5 of having the most GS profiles. In recent years, however, the predominance of Anglo-Saxon countries in the total pool of profiles has decreased from about 70% in the 1980s to around 55% in 2010 (i.e., their relative predominance decreased by 21%), when this development stagnated (see S4 Fig in S1 File for a graphical depiction). One important explanation for this decrease in predominance is that the percentage of the world's population that live in these countries also decreased from 7.3% in 1980 to 6.0% in 2018 (i.e., their relative population predominance decreased by 18%; https://data.worldbank.org).

**Change over time in sample composition.** We checked the number of researcher profiles that were cited in any particular year. This number showed an exponential increase from 1980 (the earliest year for which citations were tracked) and 2015, after which it leveled off. This indicates an increasing popularity of GS as a way to organize and track one's citations [27]. Especially during its most rapid growth between 2000 and 2010, many novel profiles have been added to the portal, as indicated by the relative lack of increase in researchers' average career age. After 2010, the average career age of profiles started increasing again, and is currently at 13.9 years (i.e., around a mid-career level). GS can thus be considered a good source of comparing the careers of scientists across a wide range of academic career stages.

## Results

### Preliminary investigation of sampling coverage

As stated in the introduction, one important limitation of our approach is that it restricts our sample to researchers who a) have a GS profile and b) describe their research focus in the

profile, and that c) assigns them to a discipline only if they used a relatively narrow discipline label to describe their focus. To evaluate the effect of these restrictions, we extracted the lists of (associate) editors all journals dedicated solely to personality psychology (and not also to social psychology or any other discipline or topic): *Journal of Individual Differences*, *Journal of Personality*, *Personality and Individual Differences*, *European Journal of Personality*, the *Personality Processes and Individual Differences* section of the *Journal of Personality and Social Psychology*, the *Journal of Research in Personality*, and *Personality Science*. On the websites of these journals' editorial boards, we found 102 unique names (see Appendix A in S1 File). Of these individuals, 90 (88%) had a Google Scholar profile, and 81 (79%) used keywords on their profile. Of those 81 people, 28 (35%) used "personality" as a keyword, 10 (12%) used "individual differences", and 9 (11%) used "personality psychology. This unsystematic search illustrates that our approach was able to identify a non-trivial but relatively low percentage of relevant profiles. We will elaborate on the limitations of this coverage rate in the Discussion and also repeated the analyses using "personality" and "individual differences" as additional keywords. For now, we proceeded with the analyses under the (seemingly reasonable) assumption that psychological disciplines would not systematically differ in the coverage rate. Still, it needs to be kept in mind that our results underestimate the actual figures, and that their greatest value therefore lie in the *relative* comparison between disciplines and over time, which we focus on in our study.

As a second check, we sampled all the journals of the 120 personality researchers. We conducted this analysis more than a year after crawling the original data collection. In the meanwhile, one researcher no longer had a GS profile, for two researchers we could not identity publications that fit our criteria (e.g., cited at least 10 times), and the publications of 7 researchers were primarily published in foreign languages, which we determined via the R-package textcat [28]. We then searched each publication outlet for occurrence of either "personality" or "individual differences" and computed the average percentage of such disciplinary publications for each of the 110 researchers. On average, 34.6% of the outlets in which the self-identified personality researchers published contained a corresponding keyword. In all but seven cases, at least one publication appeared in a corresponding outlet. This supports the validity of using self-identified labels to assign researchers to disciplines, but the imperfect overlap also highlights the fact that a focus on self-identification produces substantially different results than a focus on journals as a way to classify scholarly work in disciplines. We return to this issue in the Discussion.

### Research Question 1: Relative endorsement and international representation of disciplines

As a first step, we analyzed the degree of cross-identification across the 10 disciplines in Google Scholar. We thus established that the average percentage across disciplines was 15%. That is, the "average" discipline consisted of six out of seven members who only endorsed that discipline, and 1 out of seven members who also endorsed at least one other discipline. This number is somewhat misleading, however, because the percentage was lower in the larger disciplines, and the average cross-discipline percentage was also inflated by profiles that endorsed more than two disciplines (see the note to Table 1). On average, actually only 5% of all profiles endorsed more than one discipline. Note that these profiles were moved to a new and manually computed category of "multidisciplinary psychology" to ensure that every researcher profile was only assigned to a single discipline. This also avoided the potential bias that citations in one discipline would also count for citations in another psychological discipline, which might have biased our comparisons. As can be seen in Table 1, disciplines varied

widely in the degree of their multidisciplinarity. Whereas cognitive neuroscience, psychometrics, social psychology, clinical psychology, and psychoanalysis scored below 10%, personality psychology was an outlier in the other direction, with 43% of the researchers in this discipline endorsing at least one other domain. Further analysis indicated that 38 out of 52 multidisciplinary personality psychologists endorsed social psychology, but the remaining 14 researchers were dispersed across all other disciplines. This finding mirrors that in some countries (e.g., the US) social and personality psychology are thematically and institutionally concatenated [17]. It should be noted, however, that results for personality psychology were somewhat attenuated if the more expansive set of keywords was used (see S1 File). Specifically, the expanded set resulted in "only" 22% multidisciplinary profiles, which is still relatively high but comparable with other disciplines, such as experimental psychology.

Another aspect of Research Question 1 was to investigate relative shifts in disciplinary endorsement over historical time. We analyzed this question by first aggregating relative frequencies of all disciplines for each year between 1980 and 2018. We then checked in a multilevel analysis for between-discipline differences by testing the significance of the interaction between discipline and year in a long data format (average endorsements of each discipline nested within each year between 1980 and 2018) and found a highly significant interaction effect, $\chi^2$ ($df$ = 10) = 817.04, $p < .001$. We plotted the interaction in Fig 1 and found striking patterns. Psychometrics and cognitive neuroscience showed a strong increase over time. Also, the proportion of researchers endorsing multiple disciplines increased over time. Developmental psychology also declined somewhat around the year 2000. In each of these cases, the increases seem to have plateaued in recent years. In contrast, social psychology and psychoanalysis showed a marked decline but without clear signs (yet) of the decline leveling off. Not much historical change was found for the narrow set of personality psychology profiles. However, when the expanded three-keyword set was used, the prominence of personality psychology first decreased until the mid-nineties, after which it has been slowly but steadily increasing again (see S1 Fig in S1 File).

## Research Question 2: Topical coverage of psychological disciplines

For Research Question 2, we looked at the frequency of endorsed topic labels across profiles. Whereas all profiles had, by definition, at least one label (i.e., the label of the discipline), 84% of profiles had at least two labels, 72% had at least three labels, 51% had at least four labels, and 30% had the maximum of five labels. A preliminary analysis indicated that some frequent labels were redundant (e.g., *psychotherapy research* vs. *psychotherapy*), so we collapsed across them. Subsequently, we identified the top 10 most endorsed topics for a) psychology in general, b) each discipline in particular, and c) multidisciplinary profiles. As can be seen in Table 2, results indicated a wide variety of topics. For psychology as a whole, the ten most common topics were (in decreasing order): *emotion*, *neuroimaging*, *health psychology*, *memory*, *attention*, *social cognition*, *judgment and decision-making*, *personality*, *fMRI*, and *statistics*. Nevertheless, even these more frequent topics were only endorsed by between 3.1% (*emotion*) and 1.4% (*statistics*) of profiles, respectively.

In Table 2, the 10 most frequent topics are also listed for each discipline. As can be seen, this produced a face-valid "topic profile" for each discipline. Interestingly, there were stark differences between disciplines in the distribution of topics, with only some disciplines having a clear "signature" topic (defined as being endorsed by at least 10% of profiles within that discipline). Specifically, and not surprisingly, *psychotherapy* emerged as signature topic for psychoanalysis, *emotion* for psychophysiology, and *statistics* for psychometrics. Particularly the topics *emotion*, *health psychology*, and *personality* were highly multidisciplinary, as evidenced by

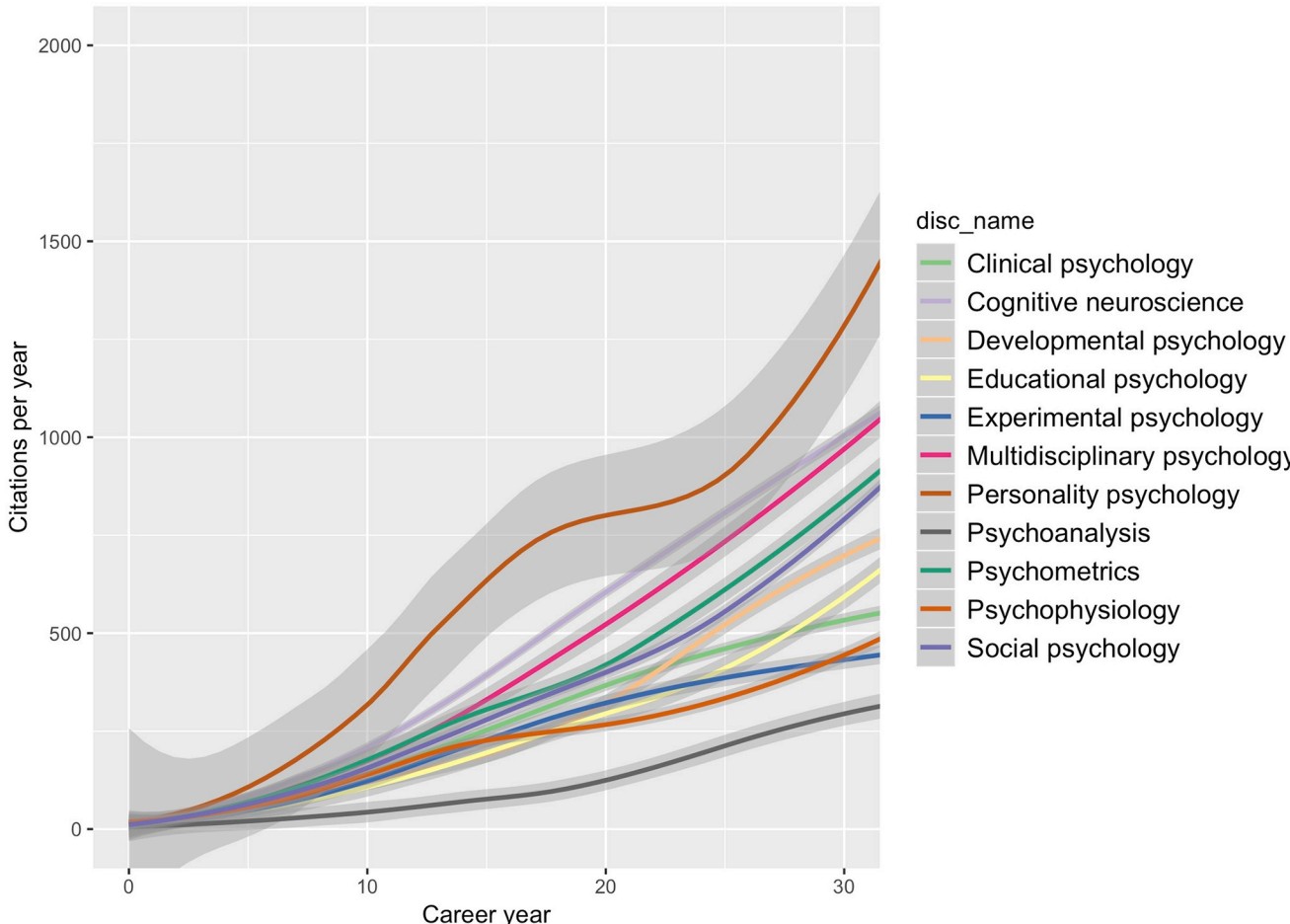

**Fig 1. Distribution of endorsements of psychological disciplines across historical time.** Endorsement proportion (y-axis) is the fraction of all profiles that endorse a certain discipline.

their prominence in many top 10 lists, including the one from researchers with a multidisciplinary profile. Profiles focusing on *psychopathology* and *psychotherapy* also appeared relatively often in the lists of most frequent topics.

We further proceeded to create a list with all 487 topics that had been endorsed at least 5 times (excluding the discipline labels), with columns indicating the absolute frequencies of endorsement within each discipline (log-transformed to reduce their skew). The co-occurrence of the vectors indicating relative topic endorsement in GS profiles across the 11 discipline columns can be expressed as a correlation matrix (487 topic frequencies × 11 disciplines). To examine whether this correlation matrix can be reduced to a smaller subset of "meta-disciplines", we conducted a factor analysis. Specifically, we ran a parallel analysis that indicated three factors, which was consistent with the visual inspection of a scree-plot. However, the three-factor solution produced an isolated factor with only one substantial loading greater than .40 (for clinical psychology). Because of this reason, we instead extracted two factors using principal axis factoring. Because there were several topics that were endorsed by multiple disciplines, orthogonality between disciplines would be an untenable assumption, so we applied an oblimin rotation.

**Table 2. Top 10 frequently endorsed topics across disciplines.**

| Rank | All profiles | Psychoanalysis | Clinical psychology | Cognitive neuroscience | Developmental psychology | Educational psychology | Experimental psychology | Personality psychology | Psychophysiology | Psychometrics | Social psychology | Multidisciplinary profiles |
|---|---|---|---|---|---|---|---|---|---|---|---|---|
| #1 | **emotion (3.1%)** | psychotherapy (10.5%) | psychotherapy (6.8%) | neuroimaging (6.7%) | cognitive_development (3.7%) | motivation (6.2%) | attention (6.5%) | **personality (6.7%)** | **emotion (16.6%)** | statistics (11.3%) | political_psychology (5.1%) | **emotion (4.6%)** |
| #2 | neuroimaging (2.3%) | psychiatry (6.8%) | **health_psychology (5%)** | memory (6.4%) | cognitive_science (3.3%) | educational_technology (6%) | cognitive_science (6.5%) | evolutionary_psychology (5.8%) | stress (4.9%) | **personality (7%)** | social_cognition (4.8%) | **personality (3.4%)** |
| #3 | **health_psychology (2.2%)** | philosophy (4.2%) | depression (3.3%) | attention (5.7%) | social_cognition (3.3%) | education (3.8%) | visual_perception (4.3%) | close_relationships (5%) | emotion_regulation (4.3%) | item_response_theory (6.2%) | intergroup_relations (4.4%) | **health_psychology (3%)** |
| #4 | memory (2.2%) | continental_philosophy (3.2%) | mental_health (3.1%) | fmri (4.6%) | evolutionary_psychology (2.8%) | learning_sciences (3.8%) | *eye_movements* (3.9%) | cross_cultural_psychology (5%) | anxiety (4%) | assessment (5.4%) | **health_psychology (4.2%)** | psychotherapy (2.7%) |
| #5 | attention (2%) | critical_theory (2.6%) | anxiety (2.9%) | aging (3.3%) | education (2.5%) | teacher_education (3.3%) | **emotion (3.5%)** | **health_psychology (5%)** | psychopathology (4%) | measurement (4.6%) | **emotion (4.1%)** | evolutionary_psychology (2.4%) |
| #6 | social_cognition (1.9%) | **emotion (2.6%)** | addiction (2.7%) | computational_neuroscience (3.2%) | adolescence (2.2%) | learning_and_instruction (2.6%) | perception (3.5%) | intelligence (5%) | eeg (3.7%) | methodology (3.9%) | prejudice (3.1%) | psychopathology (2.4%) |
| #7 | judgment_and_decision_making (1.9%) | marxism (2.6%) | psychiatry (2.7%) | judgment_and_decision_making (3.1%) | comparative_psychology (2.2%) | self_regulated_learning (2.4%) | psychophysics (3.5%) | positive_psychology (5%) | depression (3.4%) | structural_equation_modeling (3.8%) | cultural_psychology (3.1%) | social_neuroscience (2.4%) |
| #8 | **personality (1.9%)** | psychopathology (2.6%) | psychopathology (2.7%) | perception (3%) | parenting (2.1%) | learning (1.9%) | cognition (3%) | assessment (4.2%) | **personality (3.1%)** | intelligence (3.3%) | judgment_and_decision_making (3%) | attention (2.1%) |
| #9 | fmri (1.5%) | trauma (2.6%) | eating_disorders (2.4%) | eeg (2.9%) | child_development (1.9%) | metacognition (1.9%) | memory (3%) | personality_development (4.2%) | **health_psychology (2.9%)** | biostatistics (2.8%) | gender (2.7%) | social_cognition (2.1%) |
| #10 | statistics (1.4%) | cultural_studies (2.1%) | cognitive_behavior_therapy (2.3%) | language (2.7%) | autism (1.8%) | assessment (1.7%) | psycholinguistics (3%) | **emotion (3.3%)** | neuroimaging (2.9%) | quantitative_psychology (2.6%) | evolutionary_psychology (2.6%) | close_relationships (1.8%) |

*Note.* For each discipline (columns), topics are arranged in descending order of nomination frequency.
Bold topics (*emotion, health psychology, and personality*) appear across multiple disciplines.

**Table 3. Factor loadings across disciplines.**

| Discipline | "Correlational" | "Experimental" |
|---|---|---|
| Personality psychology | **.69** | -.02 |
| Clinical psychology | **.60** | .03 |
| Psychophysiology | **.54** | .28 |
| Psychometrics | **.53** | -.07 |
| Social psychology | **.49** | -.26 |
| Developmental psychology | **.48** | .11 |
| Psychoanalysis | .37 | -.10 |
| Educational psychology | .33 | -.07 |
| Cognitive neuroscience | -.01 | **1.00** |
| Experimental psychology | .17 | **.47** |

*Note.* Factor loadings from principal axis factoring with oblimin rotation are sorted on both factors in descending order. Loadings $> .40$ are displayed in bold.

Inspection of factor loadings, as presented in Table 3, indicated that the first factor was dominated by cognitive neuroscience and experimental psychology, whereas the second factor was dominated by personality psychology and clinical psychology (with substantial loadings also for psychophysiology, psychometrics, social psychology, and developmental psychology). However, some disciplines (particularly educational psychology and psychoanalysis) were not well covered by this two-factor solution. The corresponding factor solution might be qualified as "weak", yet this is expected because a stronger solution would invalidate the existence of separate disciplines. As can be seen in S3 Table of S1 File, the factor solution was basically the same when using an expanded set of personality keywords.

### Research Question 3: Publication patterns of psychological disciplines

To address Research Question 3, we first compared the average output per discipline (i.e., productivity). An ANOVA with discipline as a factor produced a highly significant difference, $F_{(10, 6.442)} = 11.31$, $p < .001$, $\eta^2 = 0.02$. In Fig 2, we plotted these differences as well as the overall distribution of productivity. As can be seen by the cumulation of data points at the lower end of the distribution, the productivity distribution resembled a power law distribution, with most researchers publishing less than 5 papers per year, but a smaller number of researchers publishing (much) more. Mean and median productivity across disciplines are displayed in Table 4. From this analysis, it emerged that cognitive neuroscience, developmental psychology, and social psychology were relatively low in productivity. By comparison, personality psychology, psychometrics, multidisciplinary psychology, psychophysiology, and clinical psychology had higher productivity levels. As can be seen in S4 Table of S1 File, productivity of personality psychologists were more in line with the average when using the expanded keyword set to identity them.

We finally compared disciplines in their ability to attract citations as a function of career progression and productivity. To begin, we ran a multilevel regression model with main effects of career year and discipline, and compared the fit with a model that additionally included their interaction. From this analysis, it turned out that the interaction was highly statistically significant, $\chi^2$ ($df = 10$) $= 2229.20$, $p < .001$. In the regression analysis, the interaction between a continuous variable (career year) and a categorical variable (discipline) is technically handled by converting the categorical variable into a series of dummy contrasts that indicate how the

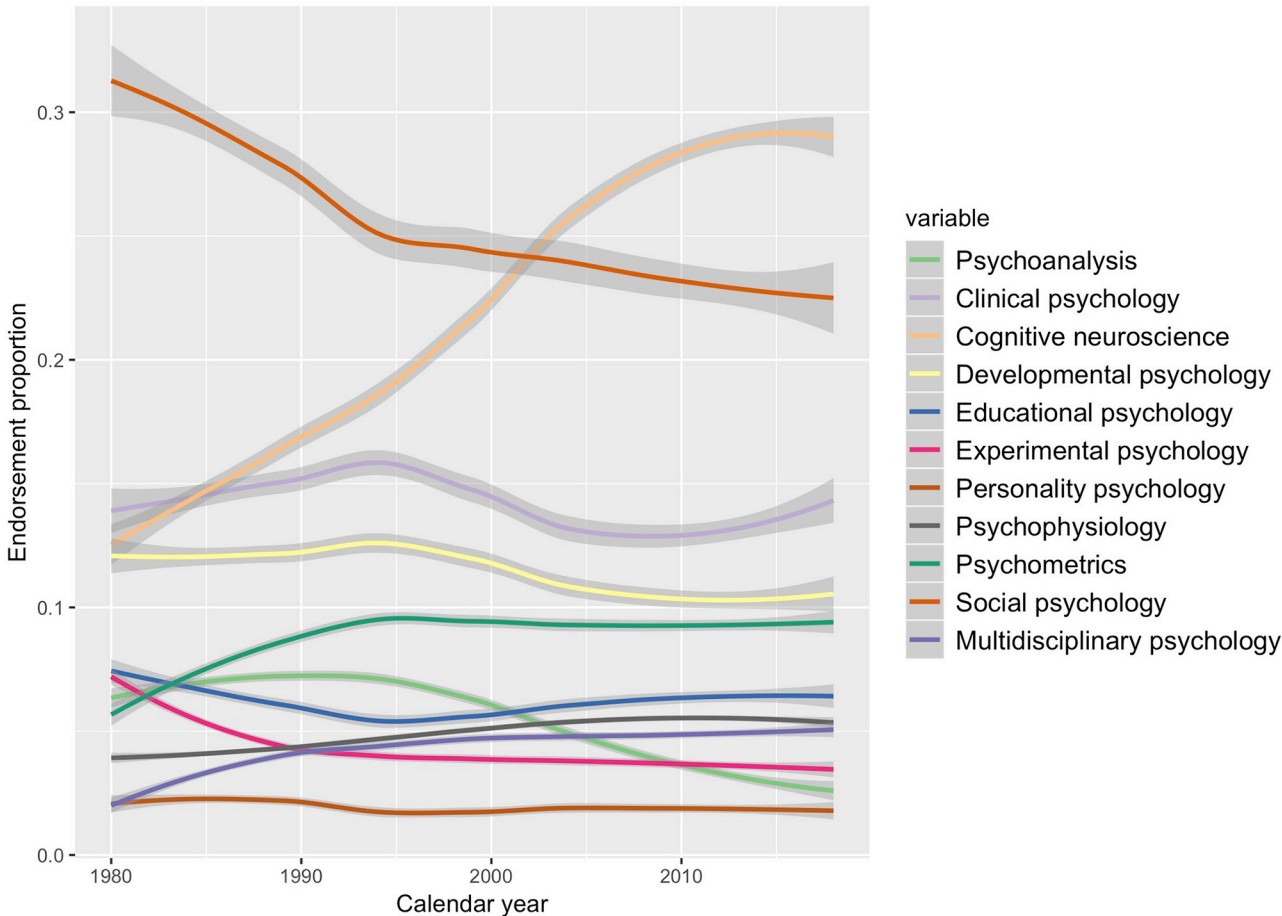

**Fig 2. Distribution of productivity (output per year) across disciplines.** Distributions for each discipline are displayed as violin plots with superimposed raw data. The red shapes indicate the mean plus 95% error bars.

**Table 4. Differences in publication productivity per year between disciplines.**

| Discipline | *M* | Median | *SD* | Min | Max |
|---|---|---|---|---|---|
| Psychoanalysis | 5.81 | 3.50 | 7.61 | 0.05 | 40.00 |
| Clinical psychology | 6.07 | 4.46 | 5.80 | 0.10 | 40.00 |
| Cognitive neuroscience | 4.99 | 3.70 | 4.45 | 0.25 | 40.00 |
| Developmental psychology | 4.61 | 3.67 | 4.19 | 0.08 | 40.00 |
| Educational psychology | 5.91 | 4.28 | 5.79 | 0.11 | 40.00 |
| Experimental psychology | 5.35 | 4.14 | 5.01 | 0.87 | 40.00 |
| Multidisciplinary psychology | 6.33 | 4.50 | 5.67 | 0.33 | 40.00 |
| Personality psychology | 6.55 | 5.46 | 5.04 | 1.00 | 25.12 |
| Psychophysiology | 5.83 | 4.50 | 5.06 | 0.50 | 40.00 |
| Psychometrics | 6.58 | 5.00 | 5.32 | 0.21 | 40.00 |
| Social psychology | 4.76 | 3.57 | 4.59 | 0.18 | 40.00 |

*Note.* Productivity describes papers published per year (appearing in Google Scholar profiles). Maximal productivity (Max) was capped at 40, which was necessary for all disciplines except for personality psychology.

corresponding slope of each factor level differs from the slope of the reference category. In the present case, we chose psychoanalysis as the reference category because this was the discipline with the lowest impact in JCR. The slope for each discipline, corresponding to the increase in citations per career year of its members, thus consists of the reference slope plus the discipline-specific interaction effect. For validation purposes, we compared these estimates with the discipline-specific aggregate impact figures from the 2019 JCR. We show the results in Table 5. Although this analysis was only based on 10 cases (personality psychology is not a separate category in JCR), the correlation was .59, $p$ = .04 (one-sided). This strengthened our faith in our regression-based approach.

As can be seen in Table 5 and Fig 3, personality psychology was the discipline with the greatest citation increase, with cognitive neuroscience, multidisciplinary psychology, psychometrics, and social psychology also attracting many citations. In contrast, the impact of psychoanalysis, clinical psychology, experimental psychology, and psychophysiology was much less. Unexpectedly, as can be seen in Fig 3, the smoothed line for personality psychology had a wider confidence interval as the other disciplines, an issue to which we will return in the Discussion section. When replicating these analyses with the expanded personality psychology keyword set, however, this discipline still ended up in a high position but not substantially different from the impact of cognitive neuroscience and multidisciplinary psychology. As can be

**Table 5. Comparison of impact indicators across disciplines.**

| Discipline | JCR domain | Predictors | | | | |
|---|---|---|---|---|---|---|
| | | JCR Median IF | JCR Aggregate IF | Career x discipline | Career (centered) × discipline | Career × productivity × discipline |
| Psychoanalysis | Psychology, psychoanalysis | 0.40 | 0.46 | 8.44 | 8.34 | 2.34 |
| Clinical psychology | Psychology, clinical | 1.93 | 2.66 | 23.51 | 24.75 | 3.87 |
| Cognitive neuroscience | Neurosciences | 3.05 | 4.02 | 36.79 | 36.92 | 6.03 |
| Developmental psychology | Psychology, developmental | 1.87 | 2.67 | 25.26 | 25.75 | 9.44 |
| Educational psychology | Psychology, educational | 1.42 | 1.91 | 24.48 | 25.58 | 3.75 |
| Experimental psychology | Psychology, experimental | 1.87 | 2.61 | 18.14 | 18.99 | 1.70 |
| Multidisciplinary psychology | Psychology, multidisciplinary | 1.32 | 2.29 | 37.83 | 36.95 | 5.08 |
| Personality psychology | NA | NA | NA | 70.50 | 72.09 | 17.43 |
| Psychophysiology | Psychology, biological | 2.18 | 2.55 | 21.33 | 21.92 | 3.14 |
| Psychometrics | Psychology, mathematical | 1.66 | 2.39 | 35.74 | 34.85 | 5.30 |
| Social psychology | Psychology, social | 1.62 | 2.08 | 35.15 | 34.69 | 6.28 |

*Note.* Career × discipline is the regression coefficient of a regression that predicts individual researchers' citations by their career age, their discipline, and the interaction between both variables. This results in discipline-specific beta coefficients, which are displayed in the table. For the Career (centered) × discipline coefficients, the interaction is based on (within-person) centered career age. Finally, the Career × productivity × discipline coefficient pertains to the three-way interaction between career age, discipline, and productivity.

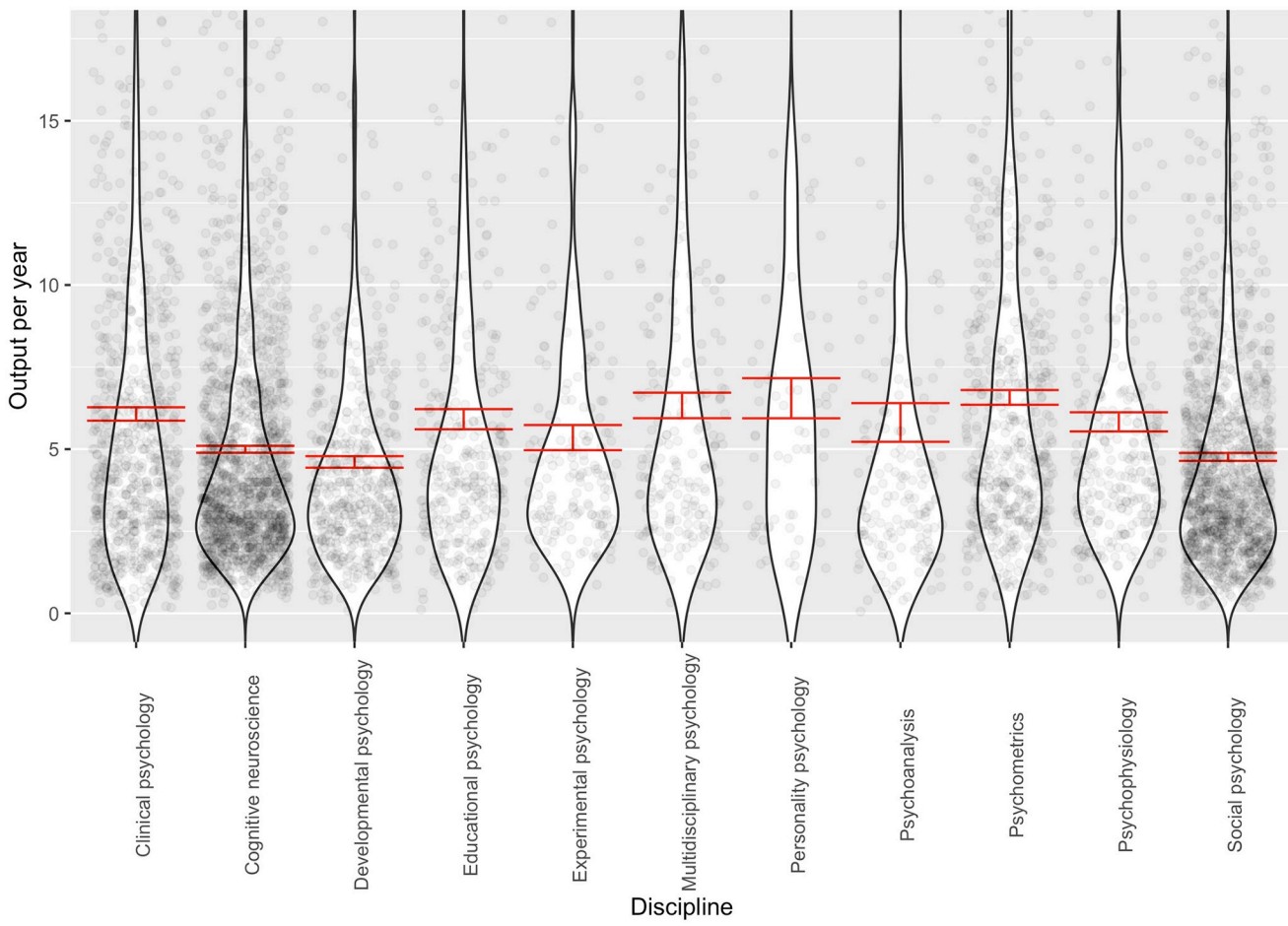

**Fig 3. Development of citations across disciplines.**

seen in S3 Fig of S1 File, the confidence intervals of the smoothed line describing the development of yearly citation volume by academic age also became more comparable with other disciplines.

To check the robustness of the results, we also ran an additional analysis with career year as a within-person centered variable to account for the possibility that differences in career lengths across disciplines might account for the results. This was not the case, however, as the pattern of results was almost identical. Furthermore, because of between-discipline differences in productivity, we wanted to investigate the interaction between career year, discipline, and productivity to estimate citation increases relative to one unit of productivity. Also for this analysis, the pattern of results was strikingly similar with the exception of developmental psychology, which emerged in a much stronger position, likely because its impact was adjusted upwards in light of its relatively low productivity. Finally, inspection of the most impactful researchers in personality psychology indicated that a Nobel prize laureate (James Heckman), who identified with personality psychology in GS, was included among them, which might have biased findings. Even after excluding Heckman from the analyses, however, personality psychology was still the most impactful discipline in all cases (although its confidence interval did overlap with some other disciplines; see S6 Table in S1 File).

## Discussion

The present study set out to conduct a comprehensive scan of psychological disciplines via GS profiles, with the goal of mapping the numerical strength of disciplines, their thematic coverage, and publication patterns. This produced an invaluable picture of psychology as a field of research, as represented in GS, which is widely used among psychological scientists today. In the following, we discuss implications for each of our three research questions.

### Relative prominence of disciplines

Our first research question concerned the numerical strength of the various disciplines as well as their co-occurrence within individual researcher profiles. Our analysis produced the striking finding that identifications with the discipline of cognitive neuroscience has seen a dramatic rise over the past 40 years, whereas those with social psychology have seen a strong decline (though the discipline has seen a large increase in absolute number of GS profiles, due to the large rise in profiles overall). In part, the relative rise in the proportion of profiles that identify with cognitive neuroscience and relative decline in the proportion of identification social psychology appear to be both sides of the same coin: If a greater proportion of researchers start to identify with cognitive neuroscience, a smaller proportion is left for identification with other disciplines (excluding endorsement of multiple disciplines, of course). This clearly did not happen with all disciplines, however. For example, the proportion of researchers identifying with mathematical psychology in their GS profiles clearly increased between 1980 and 1995. A follow-up analysis indicated that the increasing proportion of profiles from countries *without* an Anglo-Saxon background explained the decreasing proportion of profiles identifying with social psychology. Specifically, after including a variable indicating the percentage of Anglo-Saxon profiles in each year in the regression analysis, the negative association between calendar year and the proportion of social psychological profiles was no longer significant and constituted a relatively small effect size (partial $R^2 = 0.03$).

The rise of identifications with cognitive neuroscience was not necessarily surprising, as already Robins et al. [5] had reported evidence for an increasing prominence of both cognitive psychology and, to a certain extent, also neuropsychology. However, Robins and colleagues did not identify social psychology as a separate discipline, instead focusing on the school of behavioral psychology, which differs in focus from social psychology. The decline of identifications with social psychology has not appeared in previous studies and was surprising. As indicated by the additional analysis mentioned above, this decline seems mostly related to the fact that social psychology is becoming less representative of global psychology. With more and more psychological researchers joining the global research community, this might produce a different composition of psychological science in the not-so-distant future.

We also investigated differences in multidisciplinary focus across disciplines. From our scan, it emerged that only 5% of profiles endorsed more than one discipline. There were large differences between disciplines, however. Particularly the very large disciplines of cognitive neuroscience and social psychology were less frequently endorsed in combination with other disciplines. This might be seen as logical if there is more topical breadth with such large disciplines and thus perhaps less need to affiliate with other disciplines. That said, the relatively small discipline of psychoanalysis was also not characterized by many cross-disciplinary profiles, so other factors (like the degree of disciplinary identification, segregation of topics and methods) seem to also play a role here.

The large degree of multidisciplinary identification in personality psychology (although less striking when using a more expansive keyword set) is consistent with the conclusion by Yang and Chiu [15] that personality is a hub discipline with connections to and from many other

disciplines (see also [29]). However, institutional factors might also have played a role. Consistent with journals such as the *Journal of Personality and Social Psychology* as well as combined societies such as the *Society for Personality and Social Psychology*, almost three quarters of multidisciplinary personality profiles also endorsed social psychology. The remaining multidisciplinary profiles, however, endorsed combinations with all other disciplines. Finally, it should be noted that while almost 70% of all multidisciplinary personality psychologists endorsed social psychology, the reverse was not true: Only less than 3% of multidisciplinary social psychologists endorsed personality psychology, which is consistent with a recent analysis that identified personality psychology as only marginally related to social psychology [30].

As stated, only a relative minority of profiles endorsed multiple disciplines. This might be partly a result of a binary identification tendency: the idea that it is useful to firmly identify with only one discipline. This may be enforced by institutional structures, academic positions, job postings, and tenure committees that often seek (highly) specialized scholars who can represent a given field. Needless to say, however, binary classifications also have practical components given the way that psychology and academic practices are organized. After all, attending conferences and creating collaboration networks within a discipline takes time, and therefore it is much more difficult to repeat across multiple disciplines. All of this notwithstanding, we found that the percentage of multidisciplinary psychologists doubled across the study period, from ca. 2.5% to 5.0%. We can only speculate about the origins of this trend but think that three factors might play a role. First, the increase in multidisciplinarity might have intrinsic reasons. For example, researchers might be driven to study "psychology as a whole" because they really want to understand the wholeness of human functioning and perhaps realize over time that this is not possible within the confines of only one discipline. Second, there might be more recent institutional pressures towards multidisciplinarity, for example, in the tendency of large funding agencies to favor multidisciplinary work. Third, the research community itself seems to yearn more and more–especially in the wake of the replicability crisis or credibility revolution [31]–for increased cross-talk, sharing of data, and cooperation, resulting in multidisciplinary consortia and coordinated laboratories or studies.

## Topics of research

Our second research question pertained to the topics that researchers endorsed on their profiles, in addition to the disciplines they identified with. An inspection of these topics indicated that the percentage of endorsement is likely an underestimation of actual research practices within a discipline. For example, only somewhat less than 7% of clinical psychologists endorsed *psychotherapy*, and only less than 7% of cognitive neuroscientists endorsed *neuroimaging*, even though these topics appear central to the disciplines in question. This likely reflects the degrees of freedom when creating a GS profile and the fact that the choice for certain labels might limit the perceived necessity to add additional terms (e.g., developmental psychologists apparently did not deem it necessary to include terms such as *development* or *change*). As scholars can only publish 5 keywords, they need to take care in selecting keywords that are important to them, allow their easy identification (for themselves and others), and are not too redundant. Certain terms or concepts that overlap so strongly with a discipline and are implicitly contained in the discipline denomination are then likely omitted in most cases.

There were large differences between disciplines in the relative frequencies of endorsing certain labels. Overall, only psychoanalysis, psychophysiology, and psychometrics featured topics that were endorsed by more than 10% of profiles (*psychotherapy*, *emotion*, and *statistics*, respectively), whereas the relative endorsement in other disciplines was more diluted (e.g., only less than 5% of social psychologists endorsed *social cognition*). Overall, it seemed that the

larger disciplines (clinical psychology, cognitive neuroscience, and social psychology) had a somewhat stronger dilution of topics than many smaller disciplines, perhaps reflecting greater critical mass for further sub-discipline specialization. Surprisingly, especially developmental psychology seemed rather fragmented in terms of topic endorsement. Although speculative, this seems to reflect the combinatory power of the notion of development/change: Almost every psychological phenomenon changes with age/time, so a developmental psychologist can study an almost limitless array of topics. By comparison, other disciplines might be more constrained in their endorsements to certain key contents (e.g., a clinical psychologist might be more likely to study *psychopathology*).

By extracting all common topics across disciplines and counting relative endorsements of these topics per discipline, we also created "content vectors" for each discipline. By factor-analyzing these vectors, we established a novel method of mapping psychological research. Speaking to the face validity of our new method, our factor solution was reminiscent of the correlational versus experimental distinction that already Cronbach [16] has identified and that was recently confirmed by Flis and van Eck [18] using graphical mapping based on co-occurrence of terms in article abstracts.

Based on topic endorsement frequencies, we indeed found a dimension dominated by cognitive neuroscience and experimental psychology, versus a second dimension dominated by clinical psychology, developmental psychology, personality psychology, psychophysiology, psychometrics, and social psychology. Of note, these dimensions differed from the results of Yang and Chiu, who found two dimensions: basic versus applied and population-specific versus population-general. It seemed that social psychology and psychophysiology, which used to have strong experimental traditions [17], are currently focusing on topics that are also studied by traditionally "correlational" disciplines, like developmental psychology. Although speculative, it might be that more experimentally minded researchers within social psychology and psychophysiology have been increasingly gravitating towards and identifying with the upcoming discipline of cognitive neuroscience–or that new researchers coming to GS self-identify with different labels. To identify shifts over time, longitudinal research is needed on researchers' private and public self-identifications across their careers (to the best of our knowledge, this is currently not possible in GS as changes in keywords are not available to study). Additionally, the meaning of topics (e.g., *emotion*) needs to be studies across time as meanings can change and carry different connotations.

Finally, an overall inspection of topics across disciplines suggested that some topics were endorsed by more disciplines than others. Particularly three topics appeared in many top 10 lists: *Personality*, *emotion*, and *health psychology*. The relative prominence of *personality* as an overarching topic is perhaps not surprising based on earlier research that *personality* is an integrative topic studied across many disciplines [15]. Of note, however, is that *personality* was not frequently endorsed by experimentally oriented psychologists, as defined above. As a matter of fact, only *emotion* appeared also in the top 10 list of the experimental disciplines. This topic therefore seems very promising for multidisciplinary approaches and interdisciplinary integration. Indeed, experimental researchers could study the effects of emotional states on other psychological processes, social psychologists could study the effects of emotions on social outcomes, clinical psychologists could study negative emotions such as shame and depressed affect, and so forth.

## Publication patterns

As part of our third research question, we also compared psychological disciplines in terms of productivity and impact. Regarding productivity, we found that the typical (median)

psychological researcher publishes 3–4 papers per year (which are indexed in GS). Some researchers, however, publish much more than this (e.g., 11% of all researchers published 10 papers or more), thus producing a skewed distribution that corresponds to a power law. Productivity differences between disciplines were also found, with cognitive neuroscience and developmental psychology being somewhat less productive than personality psychology (only using a narrow keyword selection) and psychometrics. This might reflect the greater necessary investment in sampling in the former disciplines, with fMRI experiments and longitudinal studies being quite time-consuming to set up. By comparison, in more "productive" fields it might be more common to include additional co-authors on papers, resulting in higher numbers of papers per year.

We also compared the citation impact of the various disciplines both in terms of citation increases per year as well as in citation increases per year and publication unit (e.g., paper). Our results suggested relatively large differences between the disciplines. Results also converged with the impact statistics of the JCR, with one exception: Cognitive neuroscience did not obtain the strong citation impact that would have been predicted based on the average journal impact factors in that domain. This might not be that surprising, however, because many journals in the Neuroscience domain of Web of Science are rather medical and/or biological journals and thus from fields where impact factors tend to be higher. For cognitive neuroscientists publishing in these journals, however, research impact seems about similar to that of other psychologists, as, for example, suggested by the comparability of impact between the two largest disciplines of social psychology and cognitive neuroscience.

Psychoanalysis emerged as the discipline with the lowest impact, and also experimental and psychophysiology appeared somewhat lower in impact. Regarding psychoanalysis, the relatively low impact might reflect the earlier finding by Robins et al. [5] that this school of thought has gotten out of fashion. Consistent with this, the relative prominence of psychoanalysis visibly declined also in our analysis. Moreover, the median first year of publication for psychoanalytic profiles was markedly lower (1998) than for the other disciplines (range 2005–2007). This was not true for experimental psychology and psychophysiology, however. Because more recent technological advances, such as fMRI and neuroimaging, were less frequently endorsed by these disciplines (when compared to cognitive neuroscientists), it is possible that differences in infrastructure can explain differences in impact, but this remains speculative without further research.

Using an expanded set of personality keywords, personality psychology but also cognitive neuroscience and multidisciplinary psychology emerged as particularly high in impact. In the introduction, we speculated that interdisciplinary focus and replicability of findings might contribute to impact. In line with this, both personality psychology and cognitive psychology have been highlighted as being especially robust [21, 22]. In terms of interdisciplinary focus, the high impact of personality psychology is reminiscent of earlier claims [15, 19] that personality psychology is a hub science that attracts citations from different areas. Consistent with this, we empirically established that also multidisciplinary psychology (defined as endorsing multiple disciplines on one's GS profile) was associated with a relatively large citation impact. However, speaking against this speculation is the fact that a) the high impact of personality psychology was less evident when an expanded set of keywords was used, b) cognitive neuroscience also demonstrated strong citation impact (particularly when compared to the expanded set of personality keywords) but its members less often endorsed other disciplines, and c) experimental psychology demonstrated weaker citation impact although its members were more likely to endorse other disciplines.

In theory, the idea that multi-disciplinary research has stronger impact makes sense: If a discipline produces findings that are relevant for many other disciplines, that discipline can

accumulate more citations than a more "isolated" discipline. On a substantive level, personality psychology is concerned with various psychological variables within the "whole person" and might therefore be particularly suited to play a multi- and inter-disciplinary role. That said, it was striking that the topic of *personality* was not frequently endorsed by cognitive neuroscientists or experimental psychologists and therefore seems to primarily occupy a hub-position within correlational psychology. It is an interesting question whether there might be other, hitherto undiscovered hub-positions within experimental disciplines as well (e.g., focusing on whole-brain functioning or on interactions between psychological functions), but these were not discovered by the current analyses. It might also be the case that cognitive neuroscience itself can qualify as such a hub position within the experimental approach because it emerged as a rather strong marker of that domain in our factor analysis, as opposed to the more fragmented nature of the correlational approach. Moreover, cognitive neuroscience might be multi-disciplinary at a higher-order level, integrating knowledge from biology, medicine, engineering, and mathematics.

One interesting, unexpected finding was that the variance in impact of personality researchers in Fig 3 was much higher than the variance of other disciplines. This partly reflected the discipline's smaller size, because the pattern was not visible when a more expanded set of keywords was used (see S3 Fig in S1 File). However, the same phenomenon did not occur for the discipline of psychoanalysis, which is almost equal in size. Rather, it is possible that in personality psychology there is a relatively larger likelihood of developing an exceptionally well-cited profile, when compared to other disciplines. In other words, whereas many personality psychologists appear to follow relatively average trajectories, a sizable minority deviated from this norm and were cited many times more often. Although speculative, perhaps this pattern is due to a combination of the status of personality psychology as a hub science and its relatively small size. This combination would make it easier for clear "topic leaders" to emerge, who are then cited widely not just within personality psychology but also in other disciplines.

## Strengths and limitations

Our study had several strengths. To the best of our knowledge, we are the first to systematically scan the entire field of psychology without relying on a classification of journals. This is important because not all authors of psychology journals are psychologists, and conversely not all psychology researchers publish in psychology journals. Instead, we focused on disciplinary endorsement, which has the advantage of focusing attention on the content areas that psychological researchers themselves identify with. Using the entire scope of self-endorsed topic labels, we could therefore obtain a fuller picture of the different topics that are studied within psychology and also how they are combined. Also, a clear strength of our approach is that we included a relatively large and diverse sample of researcher profiles, which was leveraged by the fact that each profile included multiple data points per year. Using these rich data, we could compute novel impact statistics, such as citation increases per year while controlling for between-discipline differences in productivity. Finally, we employed a novel and potentially more precise index of comparative scientific impact that takes into account differences in researchers' career stage and quantitative publication output, which differs between psychological disciplines as we have found through our results.

That said, our approach also had clear limitations. Most obviously, we were limited to sampling profiles of researchers who a) created a GS profile in the first place, b) used labels to describe their research (this is not required by GS), c) used the labels that we identified as markers of each discipline, d) formulated these labels in English, and e) had more than 100 citations in GS. This clearly produced a somewhat distorted country distribution that was

skewed towards Anglo-Saxon countries, although this bias has decreased substantially in recent years. Likewise, researchers were included because they only endorsed labels that were more specific than the disciplines we used (e.g., *cognitive behavioral therapy* instead of *clinical psychology*) or did not use any label at all. The generalizability of our findings is thus limited to the degree that our GS sample is representative of the scholars of the studied fields. Within these limits, GS represents a unique possibility to sample thousands of scholars who self-identify as contributing to certain topics and fields–and links relevant information. This could barely be obtained otherwise, although we also note that future AI methods might perhaps automatically classify researchers based on keywords contained in paper abstracts.

We were able to verify that at least some of the editorial board members of mainstream personality journals indeed endorsed the corresponding discipline label in GS and also that a substantial percentage of papers of self-identified personality psychologists indeed were publishing in corresponding outlets. However, still a large number of (associate) editors did not show up in our selection of GS profiles. From our experiences with the editorial board members from personality psychology, 43 out of 81 editors who used GS labels could have been identified with a mix of 3 common keywords ("personality psychology"; "personality"; and "individual differences"). This partial success in increasing coverage might count as a "proof of principle". Moreover, by comparing topic endorsement in terms of a vector correlation, we were able to provide a first estimation of the amount of bias resulting from keyword selection. Our reported Spearman rank-order correlation of $r = .49$ between two different keywords set suggests that keyword selection did introduce method variance but our decisions were likely still valid to some extent. However, more systematic research is clearly needed that identifies for each discipline whether it is possible to identify a core set of keywords to identify most of their adherents and use this set (instead of a single keyword) for sampling purposes.

A second limitation is our selection of psychological disciplines. For example, we relied only on classic distinctions within Web of Science, supplemented with personality psychology because this is also widely seen as a core discipline. Another reason for adding personality psychology is that we are most familiar with this (relatively small) discipline, and this knowledge helped us to verify the anchor our analytic procedures and results. However, we encourage future researchers to also include additional disciplines, such as health psychology, forensic psychology, and music psychology. Furthermore, we were forced to ignore differences within disciplines. For example, within social psychology, some researchers are more focused on experimental methods, whereas others use more correlational methods [17]. The relatively static nature of our method also did not allow us to study in detail the processes by which researchers come to identify with certain disciplines, including how scientists co-construct this identification in interaction with stakeholders, like other scientists and society at large (e.g., [32, 33]).

A final limitation is our reliance on GS, which uses relatively liberal search algorithms that might not always produce valid results. For example, the year of first publication was not always computed correctly by GS, often because the author in question had a relatively common name, which sometimes led GS to claim many publications that were not actually written by the author in question (but by someone with the same or a similar name). Recently, Tang et al. [34] checked this issue for a random sample of 3,000 computer science profiles, and found that 90.5% of profiles did not contain a single publication that was falsely assigned, suggesting that the problem is relatively limited in scope (see also [35]). Still, while GS allows researchers to clean their profiles and exclude such extraneous publications, this is apparently not always done by researchers. Also, in some cases, we needed to rely on the "scholar" package's estimate of the year of first publication, which was biased (pushed forward in time) in the case of highly established researchers. Fortunately, however, we could

compute a valid indicator of first publication year by hand and established that this was associated substantially with the GS estimate, so the biasing influence seems to have been limited. Another issue with GS is that it is relatively unclear what processes (e.g., self-presentation strategies, decision rules) give rise to generating identification labels on GS. To make GS even more useful for bibliometric research, it would be helpful if it also adopted a more standardized system of label endorsement (not necessarily instead but *in addition to* the free format that is currently used).

Finally, although the average individual impact across disciplines in GS converged with the journal impact factor in JCR (which is based on Web of Science), our results should be replicated with other bibliometric platforms, such as Web of Science or Scopus. This seems currently difficult to do because other platforms do not include information about researchers' disciplinary affiliation, though there might be ways around this (e.g., automatically assigning researchers to a discipline if they publish a certain percentage of their papers in journals of any discipline). With increasing sophistication of search algorithms, it might be possible to directly compare results across platforms and discipline classification methods.

## Conclusions and implications

The current study conducted a comprehensive scan of psychological disciplines as represented in GS profiles. Our results indicated that cognitive neuroscience and social psychology have the largest number of self-identified GS profiles, but the relative composition of these profiles has shifted quite substantially from less social psychology to more cognitive neuroscience, possibly because of the latter's more prominent role outside of the Anglo-Saxon countries that used to dominate GS profiles to a much larger degree. Multidisciplinary researchers appeared a tiny, albeit increasing minority, except for personality psychology where additionally endorsing other disciplines seemed the norm. In terms of topical coverage, scientific psychology appeared focused on a variety of research themes that varies quite substantially across disciplines. Consistent with earlier conceptual and empirical analyses, the broad dimensions of correlational and experimental psychology were found to underlie the pattern of topical endorsement across the various studies disciplines. Of all possible topics, currently emotions might be seen as a potential integrating force within psychology, as it featured prominently in the thematic lists of almost all disciplines as well in the profiles of researchers with a multidisciplinary focus. It might be very much worthwhile to pursue such interdisciplinary integration, as suggested by the example of personality psychology. Specifically, personality psychology seems to represent a discipline that integrates many perspectives from other disciplines and is therefore useful for many other applied and fundamental disciplines. Institutions that want to further such integration (as well as scientific impact) might therefore be advised to focus on topics such as emotion and personality. Such an approach might also help to stem the fragmentation of academic psychology, although progress towards unification seems also contingent on a more evenly distributed focus on topics across correlational versus experimental psychological traditions.

## Supporting information

**S1 File.**
(PDF)

## Author Contributions

**Conceptualization:** Jaap J. A. Denissen, John F. Rauthmann.

**Data curation:** Jaap J. A. Denissen.

**Formal analysis:** Jaap J. A. Denissen.

**Writing – original draft:** Jaap J. A. Denissen.

**Writing – review & editing:** John F. Rauthmann.

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
