## [Decision Letter · Decision Letter 0]

26 Mar 2021

PONE-D-20-40746

A Comprehensive Scan of Psychological Disciplines Through Self-Identification on Google Scholar: Relative Endorsement, Topical Coverage, and Publication Patterns

PLOS ONE

Dear Dr. Denissen,

Thank you for submitting your manuscript to PLOS ONE. After careful consideration, we feel that it has merit but does not fully meet PLOS ONE’s publication criteria as it currently stands. Therefore, we invite you to submit a revised version of the manuscript that addresses the points raised during the review process.

I have invited two experts in the field who gave constructive comments regarding the m.s. After reading your m.s., I shared their concerns and suggestions, which could be categorized into three themes.

First is about the representativeness of the sample. Although I see that you dealt with this carefully by selecting a journal (Personality Psychology) to see how the sample may correspond to the members of the editorial board as well as the authors, the correspondence is not that high. In addition, why selecting this particular journal is not clear to the readers. Why not comparing your sample (the proportions of the sub-disciplines) with the APA division memberships? (or any other association membership information that you could find). This seems to give a better evaluation of how bias or representative the sample is. The bias of using personality psychology as an example and treating it as the bridge sub-discipline in the introduction is not grounded in current understanding. As an example, in your citation (Yang & Chiu, 2009), they grouped personality psychology with social psychology together. You treated it as if they treated personality psychology and social psychology as two sub-disciplines. The information regarding the overlap between personality psychology and social psychology in the m.s. (e.g., on p. 18) also made me wondering whether you should treat personality psychology as an independent sub-discipline if it is often overlapped with other sub-disciplines.

The second is about the clarity of the analysis. As reviewer 1 mentioned some confusion over certain analyses which I would not list here (please read the reviewer 1’s questions), I would also add that I am not sure what has been entered in the factor analysis. It reads like topics as the information is given under the topic sentence related to topics. However, because topics could be studied in various sub-disciplines, I have no idea how the findings could be indicative of certain sub-disciplines grouped together. Please revise to make sure that information regarding each analysis could be easily followed by researchers who are familiar with the given analysis. Reviewer 2’s questions also fall in this category. We are not sure how you grouped different terms/labels or made different categories.

The third is about the writing. I shared with reviewer 1’s feeling that the background of the topic is not being carefully laid out. We do not know how this research may have challenged or added to the current understanding of the field.

Overall, both reviewers and I see the strengths and merits in the manuscript. However, crucial information regarding the context and the analyses needs to be added before the m.s. could realize its full potential.

We look forward to receiving your revised manuscript.

Kind regards,

I-Ching Lee

Academic Editor

PLOS ONE

2. In line with PLOS' submission guidelines (https://journals.plos.org/plosone/s/submission-guidelines#loc-human-subjects-research), please note that outmoded terms should be changed to more current, acceptable terminology. Examples: "Anglo-Saxon" should be changed to “white” or “of [Western] European descent”, as appropriate.

Reviewers' comments:

Reviewer's Responses to Questions

**Comments to the Author**

1. Is the manuscript technically sound, and do the data support the conclusions?

Reviewer #1: Yes

Reviewer #2: Yes

2. Has the statistical analysis been performed appropriately and rigorously? 

Reviewer #1: Yes

Reviewer #2: Yes

3. Have the authors made all data underlying the findings in their manuscript fully available?

Reviewer #1: Yes

Reviewer #2: Yes

4. Is the manuscript presented in an intelligible fashion and written in standard English?

Reviewer #1: Yes

Reviewer #2: Yes

5. Review Comments to the Author

Reviewer #1: The paper presents an analysis of Google Scholar profiles of 6532 psychology researchers in terms of self-identified disciplinary affiliations. Three research questions are posed about the relative frequency of endorsement of 10 disciplines, adapted from the Web of Science classification, and changes in endorsement over time; the linkage of research topics to disciplines; and differences across disciplines in productivity and impact. Correlation and regression techniques are used to answer these questions. The authors conclude that social psychology and cognitive neuroscience are the predominant disciplines, and that the former is in decline while the latter is on the rise, possibly they suggest because researchers from non-Anglo-Saxon countries are increasing in numbers. Among their other conclusions are that research across disciplines is the exception rather than the rule; the outline of Cronbach’s two-disciplines of psychology is evident in the data; personality is a ‘hub’ discipline; and that there are possibilities for greater ‘integration’ of psychology by focusing on topics including emotion and personality.

The work is novel and interesting and adds some potential insights into how the research enterprise in psychology operates. Data analysis is careful and attempts are made to shore up interpretation where possible. On the other hand, I have some criticisms and concerns.

1. The research problem could have been better located in the existing literature. The authors rush us into their research without an account of where research on disciplinary differences is currently at and why their approach will take us forward. It is as if they are personally well-aware of the background and expect the reader to be on the same page. A fuller introduction, although it would take a little more space, would allow us to better appreciate the value of their approach and what they have been able to add through their investigation.

2. I found the analysis summarised in Table 5 difficult to follow. It has to do with comparing citation rates across disciplines. There is an explanatory note to the table and the analysis is described on pp. 21-22. I have read these several times and am still having difficulty in understanding what I can conclude from the table. For example, this statement on p. 21 is too compressed: ‘For validation purposes, we added all discipline-specific regression estimates to the reference category and compared them with the discipline-specific aggregate impact figures from the 2019 JCR.’ It would help this reader to be taken through the regression analysis more slowly, from the outcome variable to the meaning of the three-way interaction.

3. The analysis summarised in Table 3 presents the factor analysis interpreted as consistent with the two-disciplines hypothesis. The first factor has no loading of .6 or stronger and the second factor has only two defining loadings (a third is a cross loading), and this is the ‘optimal’ result after dismissing a three-factor solution. The conclusion needs to be consistent with what I would characterise as a weak factor structure.

4. The discussion is largely speculative. The authors of course must be allowed some discretion to speculate but the extent to which they engage in this makes for a loose discussion. For example, on pp. 24-25, there is a discussion of multidisciplinarity that invokes the notion of personality psychology as a hub science and offers possible reasons for an increase in the frequency of researchers engaging with multiple disciplines. Identifying a hub science on the basis that a group of researchers is less specialised is speculative rather than persuasive. Also on p. 25, the authors suggest that neuroimaging is a defining topic for cognitive neuroscience and psychotherapy a defining topic for clinical psychology. This is without substantiation and is not consistent with my understanding of those topics or disciplines. On p. 29, fashion is invoked as an explanation for the apparent decline in experimental psychology and psychophysiology, with cognitive neuroscience now the more modish label. Does this mean that experimental psychologists and psychophysiologists have not changed what they do but simply the label for self-referencing, and if so what does this mean for the validity of the conclusions that are being drawn about change?

5. My major concern is whether the methodology the authors have used is up to the weight of inference placed on it. The authors, to their credit, acknowledge limitations of their methods, both in a preliminary statement to the Results section and in the closing stages of their Discussion. These are detailed and systematic but leave the reader with real concerns about the whole approach. For example, their analysis of coverage summarised in Appendix A to me says that more than a quarter of the cases in their chosen test bed, the editors of the Journal of Personality, did not have a GS profile and only a quarter of those who did self-identified with ‘personality psychology’, which leaves me with considerable doubt about the method. I had less concern with the classification of disciplines used by the authors, although the addition of personality to the Web of Science system did allow this discipline to rise to ‘hub’ status. It is unfortunate that the authors cannot estimate in some way the bias that might have been introduced by the voluntary nature of entries in the GS data base. With so many ‘degrees of freedom’ on the researcher’s part, comparisons seem difficult to make.

6. Some minor points:

p. 4. A reference to the German Psychological Society’s position would be helpful.

P. 10, bottom of page: Has ‘Question 3’ been omitted following ‘Research’?

p. 20, l. 6. ‘greater than 1’ or .1?

Reviewer #2: This is a valuable contribution to the existing literature and I have no objections for publishing in the current form.

However, the authors may consider adding few comments concerning the following two questions. The first of which is about representative is to have a GS profile. For example, that those who identify themselves with neuroscience are more eager to create their own GS profiles than, for example, personality psychologists. Can the authors estimate approximately how many actively publishing researchers from different fields have created their personal profile in GS?

Another problem is categories. Because it is not mandatory there are many profiles, also among psychologists, who did not select any category. The next problem is the use of different labels. For example, individual_differences can be used instead personality especially in UK. Of course, there are many who used the both categories "personality" and "individual_differences" (like Lewis Goldberg), nevertheless there are many researchers who identified "individual_differences" as their research area (e.g. Frank Schmidt, Randall Engle, Thomas Bouchard, Dieter Wolke -- all with more than 30,000 citations). Perhaps the number of used categories also deserve a comment. There seems to be a difference if, for example, personality is identified as the only area or it is only one among several other research areas.

---

## [Author Response · Author response to Decision Letter 0]

16 Jul 2021

Dear Dr. I-Ching Lee,

Many thanks for allowing us the opportunity to revise MS PONE-D-20-40746, entitled, “A Comprehensive Scan of Psychological Disciplines Through Self-Identification on Google Scholar: Relative Endorsement, Topical Coverage, and Publication Patterns”.

We are very thankful for the many constructive suggestions by yourself and the reviewers. We have been as responsive as possible in addressing every suggestion. In the following, we outline for each of your suggestions how we have taken them into account.

We feel that addressing your suggestions has resulted in a much improved paper and look forward to your feedback on our revision!

Kind regards,

Jaap Denissen & John Rauthmann

Editor

1) Representativeness of the sample

1a) Although I see that you dealt with this carefully by selecting a journal (Personality Psychology) to see how the sample may correspond to the members of the editorial board as well as the authors, the correspondence is not that high.

We have acknowledged the limited overlap in sampling as a limiting factor in the discussion (p. 33) and also expanded the scope of our benchmark search to include all personality journals (see below).

1b) In addition, why selecting this particular journal is not clear to the readers. Why not comparing your sample (the proportions of the sub-disciplines) with the APA division memberships? (or any other association membership information that you could find). This seems to give a better evaluation of how bias or representative the sample is.

We are not able to use the APA Division 8 because this represents an aggregation level that is too broad (it does not include a separate personality (sub-)division). However, we did consider using membership in personality associations with which we have connections (e.g., the German Differential and Diagnostic Psychology Section, or the European Association of Personality Psychology). We decided against this because current data regulations in Europe (the so-called GDPR) do not allow us to obtain personal data (including membership information). This left us with a final possibility of sending a survey to all members and track responses. While we would certainly be interested in obtaining such information, mailing lists are quite heavily used these days, which will likely depress participation rates.

Therefore, we ultimately settled upon expanding our selection of journals, so that our results become less dependent on arbitrary sampling decisions. Specifically, we decided to sample the editorial boards of the following major journals that are solely focused on personality (JID, JoPy, PAID, EJP, JPSP:PPID, JRP, PS). On the websites of these journals, we found 102 unique names. Of these individuals, 90 (88%) had a Google Scholar profile, and 81 (79% of all editors) used keywords on their profile. Of those 81 people, 28 (35%) used “personality” as a keyword, 10 (12%) used “individual differences” and 9 (11%) used “personality psychology”. As can also be seen in the word cloud below, these three keywords were the most frequently endorsed.

As the above analysis indicates, our approach only identified a minority of personality researchers with editorial roles at dedicated personality journals. The analysis suggests that selecting the keyword “personality” would have increased our sample size. We had considered this in previous versions of the MS, but decided against this for two reasons. First, there is a difference between “personality” as a subject matter, and “personality psychology” as a discipline. While it is clear that dedicated personality editors endorse “personality” as subject matter, the reverse is not necessarily true: Many other researchers (not represented in the editorial boards of personality journals) might endorse “personality” as keyword without identifying with the discipline. Second, if expanding the keyword selection for personality psychology, we should also do this for the other disciplines, which would have delayed the project by a very long time (we would have to generate suitable keywords for all disciplines and ran all analyses with a sample size many times larger). We think this task is better left for future research and we have included a call for this in our discussion section (p. 33).

We should also note that our aim was not so much to increase coverage of the total population but to draw a representative sample of that population. The validity of our approach therefore more strongly hinges on representativity and less strongly on coverage.

1c) The bias of using personality psychology as an example and treating it as the bridge sub-discipline in the introduction is not grounded in current understanding. As an example, in your citation (Yang & Chiu, 2009), they grouped personality psychology with social psychology together. You treated it as if they treated personality psychology and social psychology as two sub-disciplines.

It is correct that the Yang and Chiu publication does did not empirically separate personality and social psychology. However, this is only because personality psychology does not have its own flagship APA journal, so JPSP is a hybrid journal. While arguing for the hub position of JPSP, Yang and Chiu specifically used arguments that were based on personality psychology as science of the “whole person” (see p. 355), and resorted to ad-hoc arguments to extend this reasoning to social psychology. We now clarify this in a footnote on p. 7. More importantly, as we stated in the introduction, personality psychology has been very clearly identified as separate core discipline and its intertwining with social psychology can be somewhat seen as a historical “accident” within US psychology that has not been repeated in Europe or elsewhere

1d) The information regarding the overlap between personality psychology and social psychology in the m.s. (e.g., on p. 18) also made me wondering whether you should treat personality psychology as an independent sub-discipline if it is often overlapped with other sub-disciplines.

As noted above, we think there are strong reasons to treat personality and social psychology as separate disciplines. What we can imagine, however, is that disciplines like personality psychology, affective science, and motivation psychology might one day merge as explicit “boundary crossing disciplines” that have a different status than the larger mono-disciplines, such as social psychology and clinical psychology. However, we think the question of (partial) redundancy is better addressed empirically, which is why we conducted factor analysis. Indeed, personality and social psychology emerged as part of a “correlational” meta-discipline, together with other disciplines. The correspondence was not so high to merit combining disciplines, however – and our results also indicated other empirical differences between social psychology and personality psychology (e.g., the latter had a somewhat higher citation impact).

2) Clarity of the analysis

I am not sure what has been entered in the factor analysis. It reads like topics as the information is given under the topic sentence related to topics. However, because topics could be studied in various sub-disciplines, I have no idea how the findings could be indicative of certain sub-disciplines grouped together. Please revise to make sure that information regarding each analysis could be easily followed by researchers who are familiar with the given analysis.

We have provided more detail about the data input for the statistical analysis (p. 20). The relevant section now reads:

“We further proceeded to create a list with all 487 topics that had been endorsed at least 5 times (excluding the discipline labels), with columns indicating the absolute frequencies of endorsement within each discipline (log-transformed to reduce their skew). The co-occurrence of the vectors indicating relative topic endorsement in GS profiles across the 11 discipline columns can be expressed as a correlation matrix (487 topic frequencies x 11 disciplines). To examine whether this correlation matrix can be reduced to a smaller subset of “meta-disciplines”, we conducted a factor analysis. “

3) The third is about the writing. I shared with reviewer 1’s feeling that the background of the topic is not being carefully laid out. We do not know how this research may have challenged or added to the current understanding of the field.

We have added a paragraph to the introduction that outlines the contribution to the field in more detail (p. 11). In the discussion, we come back to this paragraph to discuss the implications of our results.

Reviewer 1

1. The research problem could have been better located in the existing literature. The authors rush us into their research without an account of where research on disciplinary differences is currently at and why their approach will take us forward. It is as if they are personally well-aware of the background and expect the reader to be on the same page. A fuller introduction, although it would take a little more space, would allow us to better appreciate the value of their approach and what they have been able to add through their investigation.

In response to this point, we have added an elaboration of the value of our approach:

Our approach has a number of features that set it apart from the literature. First, we take a broad approach focusing on all psychological disciplines but also go into depth regarding one discipline that is often left out of analyses: personality psychology. Furthermore, we used Google Scholar to compare disciplines, which has not been done before but has a number of advantages. For example, it allowed us to flesh out the topics that each discipline tackles, and also to identify topics that are covered by multiple disciplines. Our findings can thus give rise to more constructive suggestions for topics that have the most potential for interdisciplinary collaboration. Furthermore, our method allows for the identification of linkages with the individual scholar as unit of analysis, so novel links can emerge (e.g., if topics covary in scholarly interest profiles but are typically investigated in separate papers). Our method also has important limitations, however, which we cover in the discussion section. Finally, our analysis covers a broader timespan that has featured many important developments, for example the increased globalization of academic scholarship.

2. I found the analysis summarised in Table 5 difficult to follow. It has to do with comparing citation rates across disciplines. There is an explanatory note to the table and the analysis is described on pp. 21-22. I have read these several times and am still having difficulty in understanding what I can conclude from the table. For example, this statement on p. 21 is too compressed: ‘For validation purposes, we added all discipline-specific regression estimates to the reference category and compared them with the discipline-specific aggregate impact figures from the 2019 JCR.’ It would help this reader to be taken through the regression analysis more slowly, from the outcome variable to the meaning of the three-way interaction.

We have added additional explanation to the corresponding page (p. 21).

3. The analysis summarised in Table 3 presents the factor analysis interpreted as consistent with the two-disciplines hypothesis. The first factor has no loading of .6 or stronger and the second factor has only two defining loadings (a third is a cross loading), and this is the ‘optimal’ result after dismissing a three-factor solution. The conclusion needs to be consistent with what I would characterise as a weak factor structure.

We have added the qualification of the factor solution being “weak”. We believe that such a “weak” structure is to be expected because a stronger solution would invalidate the existence of separate disciplines.

4. The discussion is largely speculative. The authors of course must be allowed some discretion to speculate but the extent to which they engage in this makes for a loose discussion.

4a) For example, on pp. 24-25, there is a discussion of multidisciplinarity that invokes the notion of personality psychology as a hub science and offers possible reasons for an increase in the frequency of researchers engaging with multiple disciplines. Identifying a hub science on the basis that a group of researchers is less specialised is speculative rather than persuasive.

We agree that we might have phrased this conclusion a bit too firmly. The conclusion by Yang and Chiu was based on citation networks and empirically demonstrated that personality/social psychology represent “hubs” in psychological citation networks. What we have added to the picture is that a) personality as a topic is endorsed by multiple disciplines, b) that personality psychologists often endorse additional disciplines, and c) they are cited more frequently. We continue to believe the overall pattern of evidence is consistent with personality being a hub science, but indeed our study only highlights part of this pattern. Moreover, cognitive neuroscience also demonstrated strong citations patterns but is less multi-disciplinary, whereas experimental psychology demonstrated weak citation patterns yet researchers in this discipline often endorsed other disciplines. We have added an acknowledgement on p. 30-31.

4b) Also on p. 25, the authors suggest that neuroimaging is a defining topic for cognitive neuroscience and psychotherapy a defining topic for clinical psychology. This is without substantiation and is not consistent with my understanding of those topics or disciplines.

We agree that this characterization was too strong and we have replaced it with: “these topics appear central to the disciplines in question.”

On p. 29, fashion is invoked as an explanation for the apparent decline in experimental psychology and psychophysiology, with cognitive neuroscience now the more modish label. Does this mean that experimental psychologists and psychophysiologists have not changed what they do but simply the label for self-referencing, and if so what does this mean for the validity of the conclusions that are being drawn about change?

We were indeed not sufficiently precise in this section. In hindsight, we think that our characterization was not valid because the relative prominence of these disciplines did not decline and it is thus not possible to claim that label endorsement shifted from these disciplines to cognitive neuroscience. Instead, we now speculate that differences in access to methods might explain the difference (p. 30).

5. My major concern is whether the methodology the authors have used is up to the weight of inference placed on it. The authors, to their credit, acknowledge limitations of their methods, both in a preliminary statement to the Results section and in the closing stages of their Discussion. These are detailed and systematic but leave the reader with real concerns about the whole approach.

5a) For example, their analysis of coverage summarised in Appendix A to me says that more than a quarter of the cases in their chosen test bed, the editors of the Journal of Personality, did not have a GS profile and only a quarter of those who did self-identified with ‘personality psychology’, which leaves me with considerable doubt about the method.

We have now included additional personality journals that boost our conclusions, but this has not changed our overall conclusion that a “correct” disciplinary self-identification on GS is by no means a universal phenomenon. We think that this cannot be expected: For various reasons, people do not have a GS profile, fail to fill out keywords, or use different (more or less specific) keywords. More systematic research of these possible biases is needed to evaluate the merits of our novel approach. However, as we now mention on p. 33, we still think our approach has value as a “proof of principle” and express the hope that our publication will invite closer scrutiny that will address some of the doubts that the reviewer expresses.

5b) I had less concern with the classification of disciplines used by the authors, although the addition of personality to the Web of Science system did allow this discipline to rise to ‘hub’ status.

In a similar vein as expressed in 5a), we now explicitly encourage that future researchers will include additional disciplines, so that the generalizability of our particular classification will be empirically testable (p. 33).

5c) It is unfortunate that the authors cannot estimate in some way the bias that might have been introduced by the voluntary nature of entries in the GS data base. With so many ‘degrees of freedom’ on the researcher’s part, comparisons seem difficult to make.

Although it falls outside of the scope of our paper, we think that it is actually possible to quantify this bias: By systematically comparing editorial board membership of discipline-specific journals, as we did for personality psychology. We now call for such a comparison in the discussion section (p. 33).

6. Some minor points:

p. 4. A reference to the German Psychological Society’s position would be helpful.

P. 10, bottom of page: Has ‘Question 3’ been omitted following ‘Research’?

p. 20, l. 6. ‘greater than 1’ or .1?

We have addressed all the minor points mentioned by the reviewer.

Reviewer 2

The authors may consider adding few comments concerning the following two questions.

1a) The first of which is about representative is to have a GS profile. For example, that those who identify themselves with neuroscience are more eager to create their own GS profiles than, for example, personality psychologists. Can the authors estimate approximately how many actively publishing researchers from different fields have created their personal profile in GS?

While a systematic comparison of “profile likelihood” across disciplines is beyond the scope of our paper (it would require us to scan the editorial boards of all discipline-specific journals in psychology), we have achieved a more accurate and representative estimate for personality psychology. We now also call for comparative research into other psychological disciplines and have toned down our conclusions in the absence of corresponding evidence (see also our responses to Points 5a-d by Reviewer 1).

1b) Another problem is categories. Because it is not mandatory there are many profiles, also among psychologists, who did not select any category.

This is correct. In our investigation of a wider sample of prominent personality researchers, we have now included a category of “GS profile without keywords”. Fortunately, our results indicated that only 12% failed to mention a profile (inverse of 88%, p. 16).

2) The next problem is the use of different labels. For example, individual_differences can be used instead personality especially in UK. Of course, there are many who used the both categories "personality" and "individual_differences" (like Lewis Goldberg), nevertheless there are many researchers who identified "individual_differences" as their research area (e.g. Frank Schmidt, Randall Engle, Thomas Bouchard, Dieter Wolke -- all with more than 30,000 citations). Perhaps the number of used categories also deserve a comment. There seems to be a difference if, for example, personality is identified as the only area or it is only one among several other research areas.

This comment is spot-on. As we explain in the above, we analyzed the profiles of (associate) editors of all dedicated personality journals and found that additionally using “personality” and “individual differences” would indeed increase coverage. By have redone all our analyses using this expanded set of labels, resulting in interesting conclusions regarding the robustness of our methods.

---

## [Decision Letter · Decision Letter 1]

2 Sep 2021

PONE-D-20-40746R1

A Comprehensive Scan of Psychological Disciplines Through Self-Identification on Google Scholar: Relative Endorsement, Topical Coverage, and Publication Patterns

PLOS ONE

Dear Dr. Denissen,

Thank you for submitting your manuscript to PLOS ONE. After careful consideration, we feel that it has merit but does not fully meet PLOS ONE’s publication criteria as it currently stands. Therefore, we invite you to submit a revised version of the manuscript that addresses the points raised during the review process.

The novelty of this paper is to investigate scholar google profiles to get some insight on the current psychology field. As the authors amply stated “Although this methodology is less suited to inform conclusions about disciplinary formation or the social organization of scientists, GS profiles do offer unique insights into the interface between individual researchers and their disciplinary identification.” The authors investigated three basic research questions: 1) their relative endorsement across time and world regions, 2) their topical coverage, and 3) publication patterns.

In the previous decision letter, I have trouble accepting the analysis of the google scholar profiles as reflecting the field because the information is of a convenient sample. I asked about the representativeness of the sample. Without information better evaluate the sample, it is better to refrain from inferring that the findings reflect the field. Unfortunately, the authors were not able to provide information to better evaluate the representativeness of the sample and in my opinion should refrain from suggesting so. Thus, to use the publications listed in the google scholar profiles to infer productivity of the subfields is reaching and contradictory to previous findings. Yang and Chiu (2007) found that “personality and social psychology is located at the heart of psychological knowledge.” You found that “researchers identifying with psychometrics being the most productive and researchers identifying with personality psychology, cognitive neuroscience, and multidisciplinary psychology as the most impactful in terms of citation increases per additional output.” I do not see how the statement could be supported if you do not have a representative sample of psychologists and detailed data regarding citations in the field.

Why? There are a lot of questions remaining in the google scholar profiles to reflect the field. As reviewer 3 mentioned, all publications (regardless of order of the authorship) are listed under the coauthors and double counting could be a problem especially if in some subfield a publication tends to have more authors than other subfields.

Also, google scholar listed publications under the same name without knowing whether it belongs to the same scholar. This could be a problem if scholars do not routinely update or check their profiles. In Yang and Chiu’s article, they target papers, not authors, and have detailed data in terms of the direction (who cites who and by comparing this directionality that they are able to infer which subfield is the hub of the psychology knowledge). It is true that in their article, personality psychology is not separated from social psychology and in their view, they do not (see p. 355-356). They wrote “…personality psychology and social psychology are intrinsically connected. They share a holistic perspective on human behaviors, integrate insights from biological and experimental psychology, suggest general principles for intervention in concrete situations, and examine the contextualized nature of basic psychological processes.” In your reply letter, you wrote that “However, this is only because personality psychology does not have its own flagship APA journal, so JPSP is a hybrid journal.” Could it be that you are implying that if there were a personality psychology journal in Yang and Chiu’s database, personality psychology, instead of social psychology, would be at the heart of psychological knowledge? If this were true, it would be more consistent with your finding that personality psychology, cognitive neuroscience, and multidisciplinary psychology as the most impactful in terms of citation increases per additional output.” However, Allik (2013) analyzed several personal psychology journals and his findings were not consistent with this interpretation. First, in Allik’s finding, “personality psychologists from the US published the largest number of papers (4924, 57.9%) and had the largest number of citations (101 875, 68.3%).” Thus, we could not simply discount the US bias because personality psychologists in the US contributed more than the half of the publications in personality psychology. In his article, he also compared those journals combined social psychology and personality and personality psychology journals, and found that personality and social psychology journals have much larger citations per papers than purely personality psychology journals. Lastly, the percentage of the papers mentioned personality was low (23.6%) of all papers in these personality and social journals. Thus, your conclusion is not consistent with the previous findings and your data do not seem to be superior to the journal publication data. As a result, please take out your current research question 3 “Publication Patterns of Psychological Disciplines” which speak to the disciplinary formation or the social organization of scientists rather than the interface between individual researchers and their disciplinary identification.

For your first question, “Research Question 1: Relative Endorsement and International Representation of Disciplines”, the research question should probably be rephrased to get at who join in the scholar google forum? In addition, I failed to see information provided for “international representation” or their subfields. As a convenient sample of psychologists, the data set is not able to tell us what the psychology as a field is about.

For your second question, in my previous letter, I have trouble understanding how you come to the classification of the topics to refer to the subfields. You entered topics (1, 0, dichotomy codes?) in factor analysis and what converged would be identified as a subfield? Shouldn’t cluster analysis or multidimensional scaling be more appropriate? Regardless, researchers named the factors, the factor analysis itself does not. Perusing the topics listed for the subfields, many topics are overlapped. What I can guess if that the topics emerged to be several factors. However, how these factors turn out to be representing subfields is still not clear to me. Please elaborate and pay attention to the overlapping topics in different topic factors. The information could be useful and not merely something to be discounted as it could refer to the interface between researchers’ disciplinary identification.

In Yang and Chiu and other’s papers on the topics, they typically found two dimensions: basic versus applied and population-specific versus population-general. How are these two dimensions correspond to your two factors? (Again, is factor analysis the best analysis strategy? Why not cluster analysis or multidimensional scaling? After all, factor analysis does not work well with dichotomous data, see Kubinger, 2003). How are the findings correspond to each other using different analysis strategies?

Lastly, even if you are trying to get information regarding the internet forum of psychologists, concerns raised by reviewer 3 are still valid and needed to be addressed.

In conclusion, I would ask you to focus your analyses and statements on the internet forum of psychologists (google scholars) and refrain from implying or stating to reflect the field. It is simply overstatement and the contradictions between your conclusions and pervious researchers do not favor you judging the quality of the data.

Kubinger, K. D. (2003). On artificial results due to using factor analysis for dichotomous variables. Psychology Science, 45, 106 – 110.

We look forward to receiving your revised manuscript.

Kind regards,

I-Ching Lee

Academic Editor

PLOS ONE

Reviewers' comments:

Reviewer's Responses to Questions

**Comments to the Author**

1. If the authors have adequately addressed your comments raised in a previous round of review and you feel that this manuscript is now acceptable for publication, you may indicate that here to bypass the “Comments to the Author” section, enter your conflict of interest statement in the “Confidential to Editor” section, and submit your "Accept" recommendation.

Reviewer #2: (No Response)

Reviewer #3: (No Response)

2. Is the manuscript technically sound, and do the data support the conclusions?

Reviewer #2: Yes

Reviewer #3: No

3. Has the statistical analysis been performed appropriately and rigorously? 

Reviewer #2: Yes

Reviewer #3: Yes

4. Have the authors made all data underlying the findings in their manuscript fully available?

Reviewer #2: Yes

Reviewer #3: Yes

5. Is the manuscript presented in an intelligible fashion and written in standard English?

Reviewer #2: Yes

Reviewer #3: Yes

6. Review Comments to the Author

Reviewer #2: This is a valuable contribution, which I would like to recommend for publishing in its present form.

Reviewer #3: This study investigated the inter-relations of different psychological disciplines and their developmental trends via a relatively new academic metric (i.e., Google Scholar Profile) compared to previous studies based on Journals, Web of Science (Journal of Citation Report), or Scopus…etc. However, using GS profile might also cause several main problems needed to resolve to make sure the results are valid and robust.

1. Although the Limitation section had pointed out that the issue of publications on Google Scholar Profile might not be written by the same author account. Therefore, it would be better if the authors can check their datasets to know how the prevalence of this issue could happen. For instance, randomly choose 20 scholars in each subdomain and investigate the proportions of this phenomenon.

2. In addition, articles with more than one author on Google scholar Profile could also raise another issue for double-counting (or even recounting several times) their citations for the same sub-domain or even in the different sub-domains. For example, if two scholars wrote an article with high citations, one scholar identified as a personality psychologist, and the other identified as a cognitive psychologist. How the authors resolve these recounting problems? Furthermore, different recounting problems might cause different meanings for the results.

3. Corresponding to the previous two comments, it would be better if the author could provide more related research articles in order to confirm that the Google Scholar profile analysis is stable and valid.

4. Another issue is about the categorization procedures of subdomains in psychology. Although “Personality” played an essential role in the whole history of psychology, based on the results of the current study, it showed the lowest Google Scholar profiles (N = 120/6532) and the largest cross-disciplinary (43%) with other domains of psychology. Therefore, the highly overlapping issue of Personality psychology with other subdomains needs further illustrations for legitimacy.

7. PLOS authors have the option to publish the peer review history of their article (what does this mean?). If published, this will include your full peer review and any attached files.

Reviewer #2: **Yes: **Jüri Allik

Reviewer #3: No

---

## [Author Response · Author response to Decision Letter 1]

18 Nov 2023

1) In the previous decision letter, I have trouble accepting the analysis of the google scholar profiles as reflecting the field because the information is of a convenient sample. I asked about the representativeness of the sample. Without information better evaluate the sample, it is better to refrain from inferring that the findings reflect the field. Unfortunately, the authors were not able to provide information to better evaluate the representativeness of the sample and in my opinion should refrain from suggesting so.

We agree that we have not drawn a representative sample of scientists and now stress this point more strongly in the discussion of our MS:

“The generalizability of our findings is [thus] limited to the degree that our GS sample is representative of the scholars of the studied fields. Within these limits, GS represents a unique possibility to sample thousands of scholars who self-identify as contributing to certain topics and fields – and links relevant information. This could barely be obtained otherwise, although we also note that future AI methods might perhaps automatically classify researchers based on keywords contained in paper abstracts.” (p. 33).

Of note, we did in fact investigate the representativeness because for the domain of personality psychology. As we report in our paper, we could improve our sampling and covered 53% of editors from major personality journals.

Thus, while our approach may not be perfect and leave open the exact boundaries of generalizability, it is one of the only and best approaches we have – so far. 

2) Thus, to use the publications listed in the google scholar profiles to infer productivity of the subfields is reaching and contradictory to previous findings. Yang and Chiu (2007) found that “personality and social psychology is located at the heart of psychological knowledge.” You found that “researchers identifying with psychometrics being the most productive and researchers identifying with personality psychology, cognitive neuroscience, and multidisciplinary psychology as the most impactful in terms of citation increases per additional output.” I do not see how the statement could be supported if you do not have a representative sample of psychologists and detailed data regarding citations in the field.

While it is true that we do not have representative sampling (we address this point above), we think that our results do, in fact, square with previous analyses that have used other methods. Specifically, also Robins identified personality psychology and neuroscience as upcoming fields in their 1999 analysis. Furthermore, Yang and Chiu also identified the combined personality/social as central to psychology. 

However, there is also a deciding difference with the Yang approach: They focused on single APA flagship journals of each discipline. However, researchers do not only publish in such journals, for example psychometric researchers also co-author publications with other researchers, for example because they ran the analyses or because they are proposing an application of a general method to solve a problem in a specific field. Our Google Scholar approach (while limited in some respects, see above) does not suffer from these limitations. We have now added a clarification about this feature of the Yang results (p. 7). 

3) Why? There are a lot of questions remaining in the google scholar profiles to reflect the field. As reviewer 3 mentioned, all publications (regardless of order of the authorship) are listed under the coauthors and double counting could be a problem especially if in some subfield a publication tends to have more authors than other subfields.

It is true that we have not accounted for authorship inflation, and we acknowledge that it can account for differences in productivity. As stated above, for example, that many methodological scholars have many co-authorships in which they provided statistical advice to primary authors (in addition to having their own publications, of course). We now acknowledge this fact in the discussion on p. 29. 

That said, we did explicitly correct for the number of produced papers in our account of scholarly impact, which indicates the added citation impact of one additional publication. We regard this as a strength of our method and have stressed this more clearly on p. 32-33.

4) Also, google scholar listed publications under the same name without knowing whether it belongs to the same scholar. This could be a problem if scholars do not routinely update or check their profiles.

This can indeed occur but fortunately the problem does not seem to be very large. Recently, Tang et al. (2021) checked this for a random sample of 3,000 computer science profiles, and found that 90.5% of profiles did not contain a single publication that was falsely assigned. Misclassification was larger for more productive researchers who identified with multiple research topics, and for researchers from China (with naming conventions being partly to blame). Results also indicated that focused only on correctly classified profiles improves the validity of rankings such as ours. We call on Google Scholar to implement better methods of disambiguation (for an overview, see Sanyal et al., 2021) and for profile owners to regularly check their profiles for incorrectly classified papers (especially if they have ambiguous names, are very productive, and/or work in different fields).

We acknowledge that this is a potential problem and have added an acknowledgement in which we also aim to quantify the likely extent of this problem (p. 35).

Identifying Mis-Configured Author Profiles on Google Scholar Using Deep Learning

Tang, J., Chen, Y., She, G., Xu, Y., Sha, K., Wang, X., ... & Hui, P. (2021). Identifying mis-configured author profiles on google scholar using deep learning. Applied Sciences, 11, 6912.

Sanyal, D. K., Bhowmick, P. K., & Das, P. P. (2021). A review of author name disambiguation techniques for the PubMed bibliographic database. Journal of Information Science, 47(2), 227-254.

5) In Yang and Chiu’s article, they target papers, not authors, and have detailed data in terms of the direction (who cites who and by comparing this directionality that they are able to infer which subfield is the hub of the psychology knowledge). It is true that in their article, personality psychology is not separated from social psychology and in their view, they do not (see p. 355-356). They wrote “…personality psychology and social psychology are intrinsically connected. They share a holistic perspective on human behaviors, integrate insights from biological and experimental psychology, suggest general principles for intervention in concrete situations, and examine the contextualized nature of basic psychological processes.” In your reply letter, you wrote that “However, this is only because personality psychology does not have its own flagship APA journal, so JPSP is a hybrid journal.”

We stand by the decision that personality and social psychology are separate disciplines. If they were intrinsically connected, they would be integrated everywhere but it is actually only the case in the USA, and perhaps some other countries (hence the decision of the American Psychological Association to merge the disciplines in one journal and collaborate in a joint social psychology/personality association, SPSP). In other countries and world regions, personality psychology is either separate (such as Germany, the UK, and Spain), or integrated with other disciplines, such as clinical psychology (this is the case in France) and developmental psychology (this is true for some universities in the Netherlands and Switzerland). We acknowledge explicitly that Yang and Chiu indeed regarded JPSP as a mixed journal (see Footnote 1), and that it requires additional argumentation to call personality a hub science (see below).

6) Could it be that you are implying that if there were a personality psychology journal in Yang and Chiu’s database, personality psychology, instead of social psychology, would be at the heart of psychological knowledge? If this were true, it would be more consistent with your finding that personality psychology, cognitive neuroscience, and multidisciplinary psychology as the most impactful in terms of citation increases per additional output.”

Indeed, we would speculate that a flagship journal dedicated to personality psychology would be more impactful. In Europe, where personality and social psychology are generally separate disciplines, it is useful to compare the European Journal of Personality (EJP) with the European Journal of Social Psychology (EJSP). As can be seen below, the impact factor of the personality journal is larger than the social psychology journal. It could be that this advantage is due to EJP also publishing review articles, whereas EJSP is limited to original research. When we search citations for non-review articles published during the past 5 years in both journals, EJSP has 6.93 citations per article, and EJP 8.74. These latter figures have a ratio of (non-review) papers from personality being 1,2612 times more cited. An even more direct comparison comes from JPSP itself. We searched the journal for the 257 articles with “personality” or “individual differences” as topic (field in WoS) vs. the 345 papers with other topics. The personality papers were cited 15.39 times per item, whereas the other (presumably social psychology) papers were cited 10.16 times per item (P:S ratio of 1.515). This corresponds to the relative advantage of personality psychology in our article. Because we think that this data point is not central to our main message, we have not included it in our revision but would be happy to add it to our supplementary materials if the editor desires so.

Table. Information from Journal Citation Reports, searched on April 21, 2022

Year

EJP

EJSP

P:S ratio

2016

3,707

1,973

1,879

2017

3,494

2,048

1,706

2018

3,329

1,775

1,875

2019

3,910

2,415

1,619

2020

5,838

3,376

1,729

7) However, Allik (2013) analyzed several personal psychology journals and his findings were not consistent with this interpretation. First, in Allik’s finding, “personality psychologists from the US published the largest number of papers (4924, 57.9%) and had the largest number of citations (101 875, 68.3%).” Thus, we could not simply discount the US bias because personality psychologists in the US contributed more than the half of the publications in personality psychology.

We are sorry about this misunderstanding. We did not want to claim that there is no separate personality psychology in the US. There are in fact many personality psychologists there, and recently there has also been the emergence of a separate US association for personality research (Association for Research in Personality, ARP). In fact, many of the researchers from our list of most impactful personality researchers were indeed affiliated with an US institutions (e.g., the top 5 personality researchers were all American: James Heckman, Richard E. Lucas, Richard W Robins, Brent W Roberts, and Todd Kashdan).

8) In his article, he also compared those journals combined social psychology and personality and personality psychology journals, and found that personality and social psychology journals have much larger citations per papers than purely personality psychology journals.

We believe this is a difficult comparison because the aggregation level of combined journals is greater than personality-only or social psychology-only journals. The more adequate comparison is the comparison between these latter two categories, and this bears out the findings reported in our paper, as described above. Moreover, please note that our initial units were GS profiles and not journals or single papers. This complicates the direct comparison between Allik’s findings and ours (they used different data and reside at different levels). Thus, even if there were a disagreement, it would not bear relevance because both sides can reflect different realities.

9) Lastly, the percentage of the papers mentioned personality was low (23.6%) of all papers in these personality and social journals. Thus, your conclusion is not consistent with the previous findings and your data do not seem to be superior to the journal publication data.

As reported above, the percentage in JPSP was higher during the last 5 years, with 42.7% mentioning “personality” or “individual differences” as a topic. Moreover, our analysis also highlighted that personality psychology is a smaller discipline than social psychology. Finally, Allik focused on articles published in dedicated journals, whereas we focused on all articles by researchers that identified with certain disciplines, which is a different level of analysis.

10) As a result, please take out your current research question 3 “Publication Patterns of Psychological Disciplines” which speak to the disciplinary formation or the social organization of scientists rather than the interface between individual researchers and their disciplinary identification.

We would like to refrain from doing so for two reasons. First, although we did not preregister this study, we did form a priori questions and hypotheses, and with these in mind we examined GS profiles and analyzed our data. Taking out a research question we deemed important to the entire picture would betray these intentions and also feel like cherrypicking. We also note that readers would likely anticipate, and even expect, this question and its findings, and the paper works better with them being integrated. Second, we are additionally convinced that we are in fact investigating the “interface between individual researchers and their disciplinary identification” because this represents the most direct interpretation of our raw data (publication data from individual researchers and their self-reported disciplinary identification). To assuage your concerns, we have reformulated several passages to be reflective of more careful and nuances thinking of the matter at hand.

11) For your first question, “Research Question 1: Relative Endorsement and International Representation of Disciplines”, the research question should probably be rephrased to get at who join in the scholar google forum? In addition, I failed to see information provided for “international representation” or their subfields. As a convenient sample of psychologists, the data set is not able to tell us what the psychology as a field is about.

We have added “(in Google Scholar)” to the description of this research question (p. 17). This should also assuage the concern that we are not able to generalize to psychology as a whole, but to the subpopulation of psychologists represented in Google Scholar. In the MS, we discuss the impact of this sampling bias at length. Further, neither we nor anyone else can quantify who joins the GS forum (that is an entirely different questions from the one we posed) because longitudinal data cannot be extracted from GS readily and there is also no easy way of identifying those not in GS. Given, however, that most researchers have GS profiles, we do not consider this research question central to understanding our Research Question 1.

12) For your second question, in my previous letter, I have trouble understanding how you come to the classification of the topics to refer to the subfields. You entered topics (1, 0, dichotomy codes?) in factor analysis and what converged would be identified as a subfield? Shouldn’t cluster analysis or multidimensional scaling be more appropriate?

At the level of the individual researcher, individuals could indeed be assigned 1 (vs. 0) if they endorsed (vs. not) a particular topic. Subsequently, we computed a fraction of researchers within each discipline who endorsed that topic (i.e., sum of all 1's divided by the number of researchers in the discipline). This resulted in a matrix in which the topics were the rows and the disciplines the columns (with the various fractions representing the relative endorsement of a topic per discipline).

13) Regardless, researchers named the factors, the factor analysis itself does not. Perusing the topics listed for the subfields, many topics are overlapped. What I can guess if that the topics emerged to be several factors. However, how these factors turn out to be representing subfields is still not clear to me. Please elaborate and pay attention to the overlapping topics in different topic factors. The information could be useful and not merely something to be discounted as it could refer to the interface between researchers’ disciplinary identification.

While it is common in factor analysis to discard “double loadings” when interpreting (and afterwards labeling) factors, you are right that topics that are represented by multiple dimensions are potentially important “interfaces” between different meta-domains of psychology. In the case of overlapping topics, this would produce a correlated factor structure, which is why we used an oblique rotation. We now highlight this more strongly on p. 20. Indeed, we highlight these overlapping topics (e.g., motivation, emotion) as a key opportunity for interdisciplinary engagement (p. 36).

14) In Yang and Chiu and other’s papers on the topics, they typically found two dimensions: basic versus applied and population-specific versus population-general. How are these two dimensions correspond to your two factors?

We appreciate these dimensions, and they indeed turned up in some of our topics (e.g., “adolescence” representing research with adolescent populations; “psychotherapy” representing an applied topic). As such, these topics had every opportunity to turn up as major organizing dimensions, yet they do not correspond well to the factors that we identified. We now comment on this in our paper (p. 28).

15) (Again, is factor analysis the best analysis strategy? Why not cluster analysis or multidimensional scaling? After all, factor analysis does not work well with dichotomous data, see Kubinger, 2003). How are the findings correspond to each other using different analysis strategies?

See above for our response why our analytical framework does in fact suit the data at hand (i.e., we used fractions instead of dummy variables).

16) Lastly, even if you are trying to get information regarding the internet forum of psychologists, concerns raised by reviewer 3 are still valid and needed to be addressed.

We have addressed the various points by Reviewer 3.

17) In conclusion, I would ask you to focus your analyses and statements on the internet forum of psychologists (google scholars) and refrain from implying or stating to reflect the field. It is simply overstatement and the contradictions between your conclusions and pervious researchers do not favor you judging the quality of the data.

We hope that our added elaborations and analyses have convinced you that our data do not point to radically different conclusions than previous researchers. That said, we happily comply with the suggestion to stress even more that our results primarily apply to the population of GS researchers, and it is an open question to what extent they might generalize to other (wider) populations.

Kubinger, K. D. (2003). On artificial results due to using factor analysis for dichotomous variables. Psychology Science, 45, 106 – 110.

Review Comments to the Author

Reviewer #2: This is a valuable contribution, which I would like to recommend for publishing in its present form.

Reviewer #3: This study investigated the inter-relations of different psychological disciplines and their developmental trends via a relatively new academic metric (i.e., Google Scholar Profile) compared to previous studies based on Journals, Web of Science (Journal of Citation Report), or Scopus…etc. However, using GS profile might also cause several main problems needed to resolve to make sure the results are valid and robust.

1. Although the Limitation section had pointed out that the issue of publications on Google Scholar Profile might not be written by the same author account. Therefore, it would be better if the authors can check their datasets to know how the prevalence of this issue could happen. For instance, randomly choose 20 scholars in each subdomain and investigate the proportions of this phenomenon.

We were actually already aware of this issue and our scripts included a check of faulty authorship, which ensured that publications not written by the focal author would be removed from the database:

 id <- id[grep(tolower(author_name), tolower(id$author)),]

Accordingly, we think that this limitations actually applies less to the present study (when it pertains to false authorship claims), but it might still pertain to researchers with very common names, whose GS profiles might attract publications from namesakes without a GS profile. This issue was empirically investigated by a recent publication by Tang (2021), and we added a corresponding reference (p. 35).

2. In addition, articles with more than one author on Google scholar Profile could also raise another issue for double-counting (or even recounting several times) their citations for the same sub-domain or even in the different sub-domains. For example, if two scholars wrote an article with high citations, one scholar identified as a personality psychologist, and the other identified as a cognitive psychologist. How the authors resolve these recounting problems? Furthermore, different recounting problems might cause different meanings for the results.

Fortunately, this is not a problem for our method because citations are tallied per author by GS (at least for the analyses reported here). Importantly, however, in case an author endorsed multiple disciplines, he or she was allocated to a “multidisciplinary” category, so comparisons between different disciplines (both in terms of quantitative output and citations) are unaffected by the mentioned problems.

3. Corresponding to the previous two comments, it would be better if the author could provide more related research articles in order to confirm that the Google Scholar profile analysis is stable and valid.

We have added recent articles that address the reliability and validity of GS profiles (Tang et al., 2021; Gasparyan et al., 2017).

4. Another issue is about the categorization procedures of subdomains in psychology. Although “Personality” played an essential role in the whole history of psychology, based on the results of the current study, it showed the lowest Google Scholar profiles (N = 120/6532) and the largest cross-disciplinary (43%) with other domains of psychology. Therefore, the highly overlapping issue of Personality psychology with other subdomains needs further illustrations for legitimacy.

We have included personality psychology a priori because it is, as the reviewer correctly pointed out, a classical domain within psychology. The fact that it emerges as a smaller domain with a large overlap with other domains (particularly social psychology) is a novel finding, which converges with its placement within the broader “social psychology” dimension in Web of Science. We think that another result reinforces its separate status, however: As a discipline, it is ideally suited to be combined with other disciplines and can thus play a role to “bridge” subdiscipline boundaries within psychology.

---

## [Decision Letter · Decision Letter 2]

13 Dec 2023

A Comprehensive Scan of Psychological Disciplines Through Self-Identification on Google Scholar: Relative Endorsement, Topical Coverage, and Publication Patterns

PONE-D-20-40746R2

Dear Dr. Denissen,

We’re pleased to inform you that your manuscript has been judged scientifically suitable for publication and will be formally accepted for publication once it meets all outstanding technical requirements.

Kind regards,

Pablo Dorta-González, Ph.D.

Academic Editor

PLOS ONE

Additional Editor Comments (optional):

Reviewers' comments:

Reviewer's Responses to Questions

**Comments to the Author**

1. If the authors have adequately addressed your comments raised in a previous round of review and you feel that this manuscript is now acceptable for publication, you may indicate that here to bypass the “Comments to the Author” section, enter your conflict of interest statement in the “Confidential to Editor” section, and submit your "Accept" recommendation.

Reviewer #1: All comments have been addressed

2. Is the manuscript technically sound, and do the data support the conclusions?

Reviewer #1: Yes

3. Has the statistical analysis been performed appropriately and rigorously? 

Reviewer #1: Yes

4. Have the authors made all data underlying the findings in their manuscript fully available?

Reviewer #1: Yes

5. Is the manuscript presented in an intelligible fashion and written in standard English?

Reviewer #1: Yes

6. Review Comments to the Author

Reviewer #1: (No Response)

7. PLOS authors have the option to publish the peer review history of their article (what does this mean?). If published, this will include your full peer review and any attached files.

Reviewer #1: **Yes: **John O'Gorman

---

## [Editor Report · Acceptance letter]

18 Dec 2023

PONE-D-20-40746R2 

PLOS ONE

Dear Dr. Denissen, 

I'm pleased to inform you that your manuscript has been deemed suitable for publication in PLOS ONE. Congratulations! Your manuscript is now being handed over to our production team.

Kind regards, 

on behalf of

Mr. Pablo Dorta-González 

Academic Editor

PLOS ONE